# Public attitudes value interpretability but prioritize accuracy in Artificial Intelligence

Anne-Marie Nussberger [1] ✉, Lan Luo[2], L. Elisa Celis[3] & M. J. Crockett [4] ✉

As Artificial Intelligence (AI) proliferates across important social institutions, many of the most powerful AI systems available are difficult to interpret for end-users and engineers alike. Here, we sought to characterize public attitudes towards AI interpretability. Across seven studies ($N = 2475$), we demonstrate robust and positive attitudes towards interpretable AI among non-experts that generalize across a variety of real-world applications and follow predictable patterns. Participants value interpretability positively across different levels of AI autonomy and accuracy, and rate interpretability as more important for AI decisions involving high stakes and scarce resources. Crucially, when AI interpretability trades off against AI accuracy, participants prioritize accuracy over interpretability under the same conditions driving positive attitudes towards interpretability in the first place: amidst high stakes and scarce resources. These attitudes could drive a proliferation of AI systems making high-impact ethical decisions that are difficult to explain and understand.

The rise of Artificial Intelligence (AI) promises unprecedented advances in many aspects of human life, including public infrastructure[1], legal systems[2], and healthcare[3]. AI systems have made great strides in learning complex patterns from large unstructured datasets, and can be used to make predictions about future outcomes. Currently available AI technologies leverage a range of methods to make predictions, from simple linear regression models to highly complex deep learning models. Simpler models are generally more interpretable, in that it is straightforward to understand why and how the AI arrives at its decisions[4]. For example, an AI system relying on linear regression might predict a health outcome from a limited set of variables (e.g., age, weight, gender) in a way that is easy to explain in simple language. More complex AI systems, such as those relying on deep learning, can be difficult or even impossible to interpret, not only to end-users, such as policy makers and citizens, but even to their engineers[4–7]. For instance, a deep learning system might predict health outcomes based on high-dimensional interactions among hundreds of variables – patterns that are impossible for human minds to grasp.

AI research initially focused on optimising AI performance, aiming to design systems that make the most accurate predictions, regardless of whether those predictions are interpretable. More recently,

however, stakeholders suggested AI interpretability is important in its own right[8]: from medical doctors rejecting the adoption of AI systems due to lack of insight about how they work[9], to business executives expressing concerns that "AI's inner workings are too opaque"[10], to social scientists proposing an "imperative of interpretable machines"[11] and the European Union establishing a right to obtain "meaningful information about the logic involved" in AI decisions[12]. These concerns have sparked debates about how much interpretability should be prioritized relative to overall AI performance, given that interpretability sometimes (but not always[13,14]) comes at the cost of accuracy[4,5,15,16]. While much of this recent attention has focused on the technical feasibility of "interpretable AI"–sometimes referred to as "explainable AI", "intelligible AI", or "transparent AI"[13,14,17]–little is known about the public's attitudes towards interpretable AI, particularly in cases where interpretability trades off with accuracy. To address this gap, we present seven empirical studies investigating whether and how much people without expertise in AI care about AI interpretability across a variety of real-world applications. We focus on characterising public attitudes towards AI interpretability rather than revealed choices, because current debates about interpretable AI take place prior to widespread technological development or deployment

[1]Center for Humans and Machines, Max Planck Institute for Human Development, Berlin, Germany. [2]Department of Marketing, Columbia Business School, New York, NY, USA. [3]Department of Statistics and Data Science, Yale University, New Haven, CT, USA. [4]Department of Psychology and University Center for Human Values, Princeton University, Princeton, NJ, USA. ✉e-mail: nussberger@mpib-berlin.mpg.de; mj.crockett@princeton.edu

of AI systems systematically varying in terms of interpretability. Hence, public attitudes–rather than revealed preferences–seem critical for policy development at present.

Because explanation is a central component of interpretable AI[4,7], psychological research on explanation provides a useful starting point for characterising public attitudes towards interpretable AI. Decades of research document explanation as a fundamental human need[18–22]. Explanations facilitate understanding and guide subsequent learning, prediction, and feelings of control[23]. In serving these functions, explanation is essential for establishing trust[18,23,24]. If explanations play a similar role in human-machine interactions, interpretability will be a necessary precondition for establishing trust in AI[7,11]. Indeed, a recent study provides initial evidence: when people perceived an AI joke-recommender system as opaque, they avoided relying on its recommendations, even when they knew that the AI outperformed humans in terms of accuracy[25].

In the current work, we investigate several factors hypothesized to drive attitudes towards interpretability in AI. Currently available AI systems can make decisions autonomously, or merely provide recommendations for human users to implement. In view of past work suggesting that people demand more explanation from intentional agents[22], we wanted to explore whether people value interpretability as more important for AI systems deciding in an autonomous capacity relative to systems providing recommendations.

Second, we predicted that people would consider interpretability as more important in settings with higher stakes (e.g., medical care, criminal justice) than in settings with lower stakes (e.g., entertainment, shopping). Because interpretability plays a crucial role for predicting, auditing, and controlling underlying decision-making processes[23,26], it should be particularly important in settings where AI has large consequences for human welfare. Considering low- versus high-stake cases within the same domain reinforces this intuition: you would probably care more about understanding why an AI accepted or rejected your application for a salaried permanent job, compared to an unpaid honorary job. Indeed, decades of research have documented that people demand explanations more for high-stakes than low-stakes decisions[7,13,23].

A third aspect of AI applications that might drive attitudes towards interpretability is its potential gatekeeping function. Many emerging AI applications are designed to determine access to scarce but desirable resources, such as jobs, financial loans, or medical care. Ample theoretical and empirical work demonstrates that people demand explanations for decisions involving the allocation of resources[27–29], especially when those resources are scarce[30,31], in order to ensure that the allocation procedure was fair. Importantly, existing literature suggests that people's concerns about fairly allocating scarce resources are dissociable from concerns about stakes[29]. This suggests that people may be more concerned about AI interpretability in applications that allocate scarce resources, independent of the stakes at hand.

Finally, because AI interpretability sometimes comes at the cost of AI accuracy, we sought to characterise people's attitudes towards interpretability as a function of accuracy and in direct tradeoffs between interpretability and accuracy. Previous work from psychology suggests people might perceive AI-accuracy as a proxy, or at least precondition, for ensuring favourable outcomes[28,31]. This could lead people to prioritize accuracy over interpretability, despite valuing interpretability in its own right.

By testing these hypotheses among non-experts (see the Methods section for summaries of participants' computer science knowledge), we sought to address the present lack of empirical insights about public attitudes towards AI interpretability. We first surveyed attitudes about the importance of interpretability across a variety of real-world applications where AI systems either made recommendations to a human decision-maker or made decisions on behalf of a human

decision-maker (Study 1A). This initial study indicated positive attitudes towards interpretability that were similarly pronounced for AI systems making decisions or recommendations and that varied substantially across applications. Two pre-registered follow-up studies with samples nationally representative for age, race, and gender in the US and UK provided further correlational evidence that stakes and scarcity predict variation in positive attitudes towards interpretability in AI (Studies 1B and 1C). Using an experimental study design, we then confirmed that stakes and scarcity have a causal impact on attitudes towards interpretability (Study 2). Next, we demonstrated that people value AI interpretability largely independently of AI accuracy (Study 3A). However, when interpretability and accuracy directly traded off, these attitudes proved capricious with participants willing to sacrifice interpretability for the sake of accuracy (Studies 3B and 3C).

## Results
### Study 1A–establishing attitudes towards AI interpretability across a variety of applications
We conducted a behavioural experiment to examine people's attitudes towards interpretability in AI across a variety of applications. Participants (final $N = 170$; US convenience sample recruited via Amazon's Mechanical Turk, MTurk) first read a definition of 'explainable AI', specifying that "by explainable we mean that an AI's decision can be explained in non-technical terms. In other words, it is possible to know and to understand how an AI arrives at its decision" (see SI Notes, Materials Study 1A). Following past work in psychology and philosophy of science[7], we used the more intuitive term 'explainable' rather than 'interpretable' while ensuring that our definition aligned with both terms' prevalent use in existing work on interpretable AI[4]. Each participant read twenty descriptions of real-world AI applications ranging from allocating medical treatment to news reporting to photo assistants. We compiled the collection of AI applications by surveying newspaper articles, technological reports, and scientific papers, with the aim of covering a diverse range of applications already in use as comprehensively as possible (see Fig. 1 for an overview; see SI Notes, Materials Study 1A for full list of applications with source links and instructional descriptions).

To explore the role of AI autonomy for people's attitudes towards the importance of interpretability, half of the participants were randomized to read a recommend version that described an AI system making recommendations to a human decision-maker, while the other half of participants read a parallel decide version that described an AI system deciding on behalf of a human user. For example, the recommend version of the 'medical treatment' application read "An AI recommends to a doctor what disease a patient might be suffering from", whereas the corresponding decide version read "An AI establishes on behalf of a doctor what disease a patient might be suffering from". Two applications ('surveillance' and 'virtual assistants') were included only as decide versions, as a parallel recommend version was not sensical. For each of the twenty applications, which were presented one by one and in randomized order, participants answered the question "how important is it that the AI in this application is explainable, even if it performs accurately?" on a discrete 5-point scale with three labels (1 = not at all important, 3 = moderately important, 5 = extremely important).

First, we examined the effect of AI autonomy (recommend versus decide) on participants' attitudes towards the importance of interpretability for those applications that existed in both a recommend and a decide version. Because participants gave their answers on a discrete rating scale, we used mixed effect ordinal regression analysis with a fixed effect for condition and a random intercept effect for participant. There was no significant difference in participants' interpretability-ratings across the two conditions, $\chi^2(1) = 1.72$, $p = 0.189$, $OR_{decide} = 1.23$, 95% $CI_{OR}$ [0.90, 1.67], $p = 0.188$. Median ratings coincided at 4 ($IQR_{recommend} = 3$, $IQR_{decide} = 2$), above the scale's

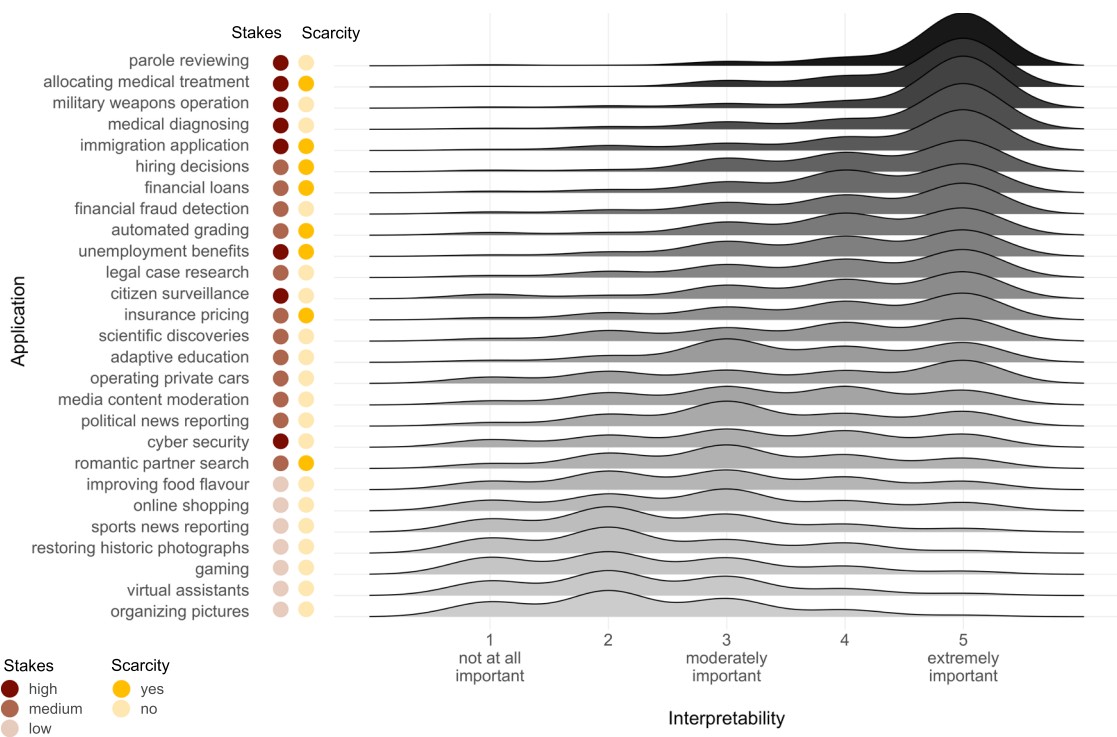

**Fig. 1 | Attitudes towards interpretability across real-world AI applications.** Joyplot visualizes the distributions of interpretability ratings, averaged across recommend and decide versions. Participants (*N* = 170) responded to the question "how important is it that the AI in this application is explainable, even if it performs accurately?" on a 5-point rating scale (1 = not at all important, 5 = extremely important).

"moderately important" midpoint. These results indicate robust and positive attitudes towards interpretable AI across a variety of applications, that seem to be largely independent of AI systems' autonomy.

However, as illustrated in Fig. 1, we also observed substantial variation in attitudes towards AI interpretability across applications. Collapsing across recommend and decide conditions, participants rated interpretability most important for applications such as 'parole reviewing' (Mdn = 5, IQR = 0), followed by applications such as 'political news reporting' (Mdn = 3, IQR = 1), and least important for 'organising pictures' (Mdn = 2, IQR = 1). Hence, our next step was to explore whether variation across AI applications in terms of the involved stakes and scarcity[7,19,30] predicted variation in attitudes towards interpretability. To this end, two of the authors performed hand-coded categorisations of the stakes (low/medium/high) and scarcity (no/yes; see Fig. 1) involved in a given application after data collection was complete. To avoid issues of multicollinearity (most applications involving scarcity involved high stakes), we ran separate ordinal mixed effect regression models to explore effects of stakes and scarcity on attitudes towards interpretability. Regressing on interpretability ratings with a fixed effect for stakes and a random intercept effect for participant, we observed a significant main effect ($\chi^2(2) = 1066.20$, $p < 0.001$) signifying that participants valued interpretability more in medium- (OR = 2.15, 95% CI$_{OR}$ [1.94, 2.37], $p < 0.001$) and high-stakes applications (OR = 3.18, 95% CI$_{OR}$ [2.93, 3.44], $p < 0.001$) relative to low-stakes ones, and more in high-stakes relative to medium-stakes ones, OR = 1.03, 95% CI$_{OR}$ [0.83, 1.24], $p < 0.001$ (Holm-correction applied for all multiple comparisons). A separate model including a fixed effect for scarcity and a random intercept effect for participant showed a significant main effect ($\chi^2(1) = 192.71$, $p < 0.001$) signifying that participants valued interpretability as more important for applications involving the allocation of scarce resources, relative to those that did not, OR = 2.68, 95% CI$_{OR}$ [2.32, 3.09], $p < 0.001$.

The results of Study 1A demonstrate overall positive attitudes towards interpretability that generalise across less autonomous AI

systems, which make recommendations, and more autonomous ones that directly make decisions on behalf of human agents. We also found exploratory evidence that the stakes and scarcity characterising a given application might explain variation in attitudes towards interpretable AI. In our next studies, we sought to replicate these exploratory findings in representative non-expert samples drawn from different populations, and to test their robustness to using a validated categorisation of stakes and scarcity as well as their robustness to varying the language used to probe attitudes towards interpretability.

## Study 1B–replicating attitudes towards AI interpretability in a representative US sample

Next, we tested whether the previous study's findings would replicate in a sample from the US (final *N* = 258) that was representative in terms of gender, age, and race and that was recruited from a different platform, Prolific Academic. We dropped the manipulation of AI autonomy (using only the decide version) and instead focused on testing whether the observed attitudes towards interpretability in AI were robust to varying the language used to probe them and to using a validated categorisation of the applications in terms of involved stakes and scarcity. In particular, we used the term "understandable" instead of "explainable" throughout the instructions and slightly changed the answer format from a dichotomous measure to a continuous slider with the same labels as in Study 1A to allow more fine-grained responses. To validate the post-hoc categorisation of applications, we had nine independent raters (i.e., who were blind to the study hypotheses) categorise each application according to the involved stakes (low/medium/high) and scarcity (no/yes). Aggregating across vignettes, raters agreed in their stakes categorisations 70% of the time and in their scarcity categorisations 84% of the time. The pre-registered procedure, hypotheses, and analysis plan are available at the Open Science Framework[32].

Following the pre-registered analysis plan, we first tested whether attitudes towards interpretability (~understandability) in AI exceeded

**It is flu season. An AI decides whether or not a citizen will get a vaccine.**

|  | High scarcity | Low scarcity |
|---|---|---|
| High stakes | In this case, **the vaccine supply is very limited**. Because the vaccine is very expensive, laborious to produce, and must be stored at low temperatures, **the vaccine cannot be stocked in large quantities.**<br><br>This particular **vaccine protects against a deadly variant of the flu.** | In this case, **the vaccine supply is abundant**. Because the vaccine is very cheap, easy to produce, and can be stored at any temperature, **the vaccine can be stocked in large quantities.**<br><br>This particular **vaccine protects against a deadly variant of the flu.** |
| Low stakes | In this case, **the vaccine supply is very limited**. Because the vaccine is very expensive, laborious to produce, and must be stored at low temperatures, **the vaccine cannot be stocked in large quantities.**<br><br>This particular **vaccine protects against a mild variant of the flu.** | In this case, **the vaccine supply is abundant**. Because the vaccine is very cheap, easy to produce, and can be stored at any temperature, **the vaccine can be stocked in large quantities.**<br><br>This particular **vaccine protects against a mild variant of the flu.** |

**Fig. 2 | Exemplary instructions from Study 2.** Schematic representation of the instructions for the vaccine application with its four versions. Each version was shown on a separate page, with the same general scenario described at the top. The depicted bolding and underlining corresponds to the format shown to participants.

the scale-midpoint "moderately important". This was the case, $M = 3.70$, SD = 1.24, $t(7,481) = 49.32$, $p < 0.001$, 95% CI [3.68, 3.73], $d = 0.57$. Deviating from the pre-registered analysis plan, we estimated separate mixed effect regression models for stakes and scarcity due to multicollinearity of the two predictors. Because participants now answered on a continuous slider scale, we used linear regression analysis with the respective fixed effects for stakes and scarcity and a random intercept effect for participant. We replicated a significant main effect for stakes, $F(2) = 2,803.30$, $p < 0.001$. Relative to applications involving low stakes, people valued interpretability more in applications involving medium ($b = 0.93$, $p < 0.001$, 95% CI [0.85, 1.00]) or high stakes ($b = 1.49$, $p < 0.001$, 95% CI [1.42, 1.55]), and more amidst high relative to medium stakes, $b = 0.56$, $p < 0.001$, 95% CI [0.50, 0.62] (Holm-correction applied for all multiple comparisons). Similarly, a significant main effect for scarcity ($F(1) = 364.86$, $p < 0.001$) indicated that people valued interpretability more in applications allocating scarce resources, relative to those that did not, $b = 0.58$, $p < 0.001$, 95% CI [0.52, 0.64].

**Study 1C – Replicating attitudes towards AI interpretability in a representative UK sample**
To further verify the robustness of our results, we ran another replication using a representative sample from the United Kingdom (final $N = 246$) recruited from Prolific Academic. We applied the same instructions and procedures as in Study 1B, as also pre-registered at the Open Science Framework[32].

Again, attitudes towards interpretability in AI exceeded the scale-midpoint "moderately important", $M = 3.68$, SD = 1.26, $t(7,133) = 46.02$, $p < 0.001$, 95% CI [3.66, 3.71], $d = 0.54$. We also replicated a significant main effect for stakes, $F(2) = 2,823.10$, $p < 0.001$. Relative to applications involving low stakes, people valued interpretability more in applications involving medium ($b = 0.95$, $p < 0.001$, 95% CI [0.88, 1.03]) or high stakes ($b = 1.52$, $p < 0.001$, 95% CI [1.45, 1.59]), and more amidst high relative to medium stakes, $b = 0.56$, $p < 0.001$, 95% CI [0.50, 0.62] (Holm-correction applied for all multiple comparisons). Similarly, a significant main effect for scarcity ($F(1) = 364.86$, $p < 0.001$) indicated that people valued interpretability more in applications allocating scarce resources, relative to those that did not, $b = 0.57$, $p < 0.001$, 95% CI [0.50, 0.63].

Across representative samples from the US and UK, Studies 1B and 1C replicated robustly positive yet variable attitudes towards interpretability in AI. Again, stakes and scarcity emerged as potential driving forces in people's valuations of interpretability. Still, these findings concerning the role of stakes and scarcity remained correlational; the applications we tested varied on a number of other dimensions; and also in the validated ranking, stakes and scarcity covaried in the sense that almost all applications involving high scarcity also involved high stakes. Indeed, there was no application involving low stakes but high scarcity in the validated ranking. Thus, we next pursued an experimental approach to test the hypothesis that stakes and scarcity independently drive attitudes towards interpretability in AI.

**Study 2–characterising attitudes towards AI interpretability: stakes and scarcity as driving forces**
To examine whether stakes and scarcity impact attitudes towards interpretable AI, we manipulated these factors in a 2 × 2 within-subjects design, focusing on five autonomous applications: allocating vaccines, prioritizing hurricane first responders, reviewing insurance claims, making hiring decisions, and prioritizing standby flight passengers. Participants (final $N = 84$; US convenience sample recruited from MTurk) were presented with the four versions of each given application in randomised order. Figure 2 illustrates how the four different versions read for the 'allocating vaccines' application:

For each application and version, participants answered the question "In this case, how important is it that the AI is explainable?" using a slider ranging from "not at all important" to "extremely important". Below the slider, we displayed a note reminding participants that "Explainable means that the AI's decision can be *explained in non-technical terms*. Please consider how important it is that the AI is explainable, *even if it performs accurately*" (emphasis from original instructions; see SI Notes, Materials Study 2).

Because our experimental manipulation implied that stakes and scarcity varied independently, we were able to run full mixed effect regression models including fixed effects for stakes and scarcity, as well as their interaction, as well as random intercept effects for participant and application. Aggregating across applications, type II Wald $\chi^2$ tests indicated significant main effects for stakes ($\chi^2(1) = 348.48$, $p < 0.001$) and scarcity ($\chi^2(1) = 110.98$, $p < 0.001$) on attitudes towards interpretability, which were not qualified by an interaction, $\chi^2(1) = 0.10$, $p = 0.754$ (Fig. 3a). In particular, participants cared more about interpretability for high- relative to low-stakes cases ($b = 0.85$, $p < 0.001$, $d = 0.33$, 95% CI [0.25, 0.40]) and for high- relative to low-scarcity cases, $b = 0.49$, $p < 0.001$, $d = 0.19$, 95% CI [0.11, 0.26]. This pattern

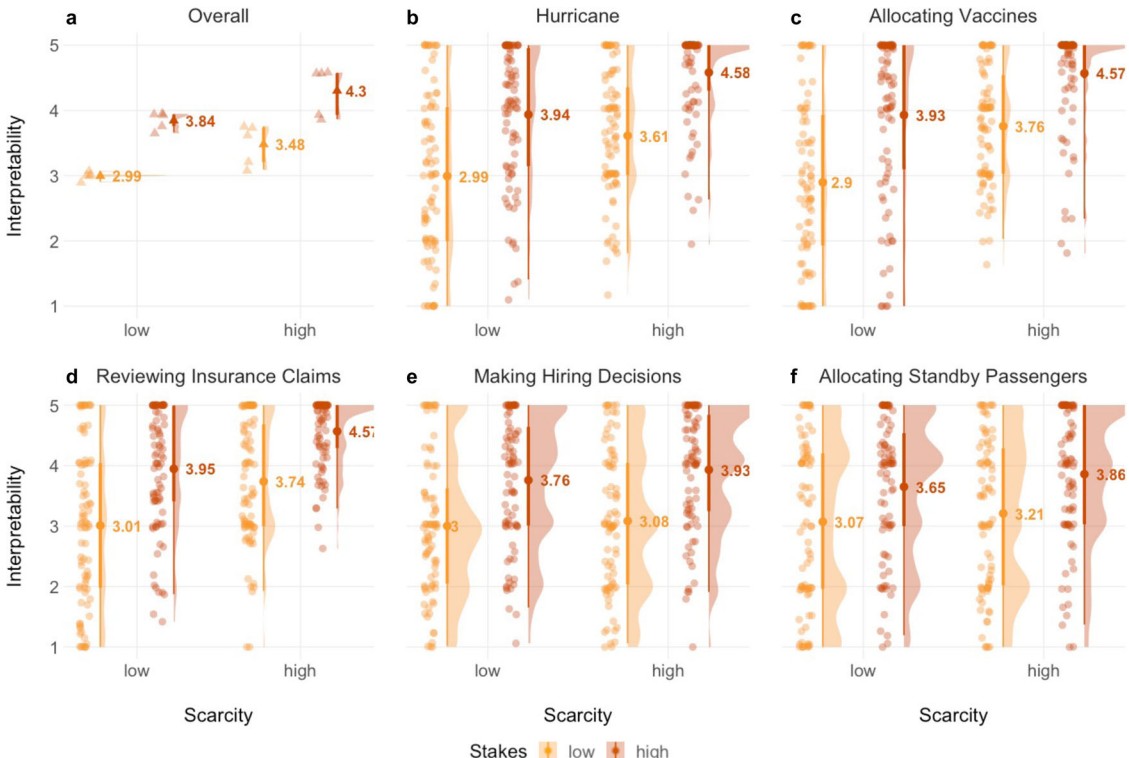

**Fig. 3 | Results for Study 2.** Participants' responses from Study 2 ($N = 84$) to the question "In this case, how important is it that the AI is explainable?" on a continuous slider-scale from "not at all important" (1) to "extremely important" (5). All panels show the jittered raw data, its density, the point estimate of the mean with its 95% confidence intervals, and interquartile ranges; all grouped by stakes (indicated by fill colour; low stakes = yellow, high stakes = red) and scarcity (indicated on x-axes). In summary, participants rated interpretability as more important for high stakes and high scarcity situations. Main effects for stakes and scarcity were not qualified by an interaction. **a** data aggregated across all five applications; triangle-shaped data points represent averages for the five applications. **b**–**f** non-aggregated data for each individual application; circle-shaped data points represent individual responses.

replicated across the five different applications (Fig. 3b–f). Overall main effects for stakes and scarcity were robust when we added gender, age, education, income, pre- and post-task support for AI, and computer science knowledge to the model (see SI Results, Study 2).

To summarize so far, our first four Studies established that people consistently value interpretability across a wide range of AI applications and that they value interpretability more when AI makes decisions involving high stakes and scarce resources. In these studies, we held the level of AI accuracy constant by explicitly instructing participants to rate interpretability's importance for a given application "even if the AI performs accurately". Because it has been widely argued that, in practice, interpretable AI may require trading off interpretability against accuracy[4,5,15,16], in Studies 3A-C we sought to investigate people's attitudes towards interpretability in AI across different levels of accuracy and when interpretability explicitly comes at the cost of accuracy.

### Study 3A–characterising attitudes towards AI interpretability as a function of accuracy

Taken together, the previous studies suggest that people hold positive attitudes towards interpretability in AI. Our instructions across these studies told participants to assume the AI would perform accurately. This raises the question whether people's attitudes towards interpretability are stable across AI models that vary in accuracy. To address this, we asked participants (final $N = 261$ recruited from Prolific; the sample was representative of the US population in terms of gender, age, and race) to indicate their attitudes towards interpretability for separate AI models that varied in their accuracy between 60% and 90%. For each of the AI applications from Study 2, participants rated the importance of interpretability on four separate sliders where each

slider represented a separate AI model performing at a specified accuracy level. We focused on the range between 60% and 90% accuracy (presented in increments of ten percentage points) because models that perform merely at chance-level or only slightly better are undesirable per se, and because few models available to date achieve accuracy levels above 90%. We counterbalanced the order in which we presented the AI models across participants (low (60%) to high (90%) for half of participants, high to low for the other half). Because we focused on characterising attitudes towards interpretability as a function of accuracy, we dropped the variations of stakes and scarcity and presented only the general description of each AI application (e.g., "It is flu season. An AI decides whether or not a citizen will get a vaccine"). The pre-registered sampling plan, procedure, and materials are available at the Open Science Framework[32].

To explore whether participants' attitudes towards AI interpretability were sensitive to variations in AI accuracy, we ran a linear mixed effect model predicting rated importance of interpretability by a fixed effect of accuracy and random intercept effects for participant and application. A type II Wald chi-square test indicated a significant effect of accuracy on interpretability importance, $\chi^2(3) = 11.89$, $p = 0.008$, such that participants rated interpretability as less important for AI models with higher accuracy both at the overall level (Fig. 4a) and across all five AI application (Fig. 4b–f). This overall pattern replicated when accounting for various control variables and in particular was not affected by the order in which we presented the AI models varying in accuracy ($p = 0.422$; see SI Results, Study 3A). Notably, across all levels of accuracy and including the 90% level, participants indicated a high level of importance for AI interpretability such that their ratings consistently exceeded the "moderately important" scale-midpoint ($M$s ≥ 3.72; one-sample $t$-tests yielding $p$s < 0.001, Cohen's $d$s ≥ 0.54).

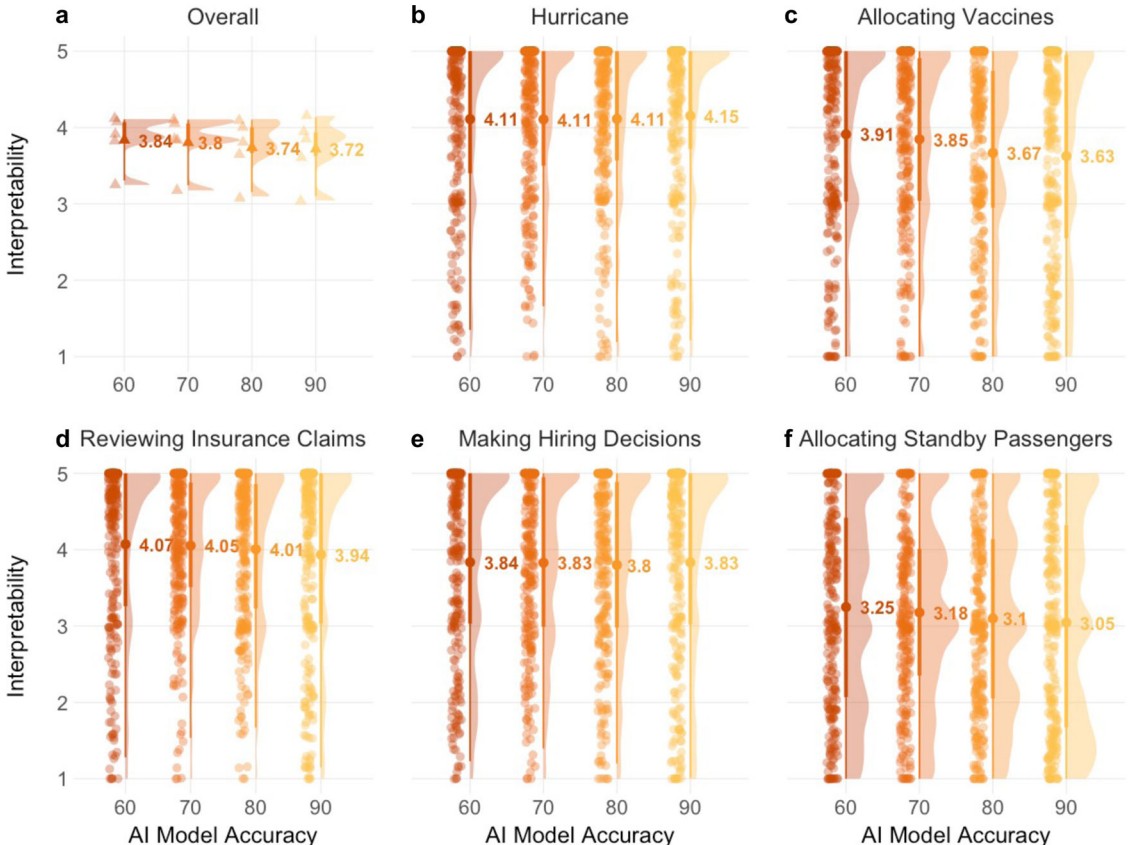

**Fig. 4 | Results for Study 3A.** Participants' responses from Study 3A ($N$ = 261) to the question "How important is it that the given AI model is explainable?" on continuous slider-scales from "not at all important" (1) to "extremely important" (5). For each AI model with a given level of accuracy, there was a separate slider-scale. Panels show the jittered raw data, its density, the point estimate of the mean with its 95% confidence intervals, and interquartile ranges. Overall, there was a slight tendency for participants to rate interpretability as less important for more accurate models. **a** Data aggregated across all five applications; triangle-shaped data points represent averages for every of the five applications. **b**–**f** Non-aggregated data for each individual application; circle-shaped data points represent individual responses.

Our findings from Study 3A indicate that attitudes towards interpretability in AI are stable across different levels of AI accuracy and that they average at a level valuing AI interpretability consistently as more than "moderately" important. While Study 3A asked participants to evaluate the importance of interpretability across independently varying levels of accuracy, in practice AI interpretability might come at the cost of AI accuracy[4,5,15,16]. Thus, in our next step we sought to explore how people value AI interpretability when it comes as a tradeoff with AI accuracy.

### Study 3B–characterising attitudes when AI interpretability trades off with AI accuracy

To examine how people value interpretability when it comes at the cost of accuracy, we presented participants (final $N$ = 112; US convenience sample recruited from MTurk) with a slider measure where one end represented a "completely accurate" but "not at all explainable" AI, whereas the other end represented a "not at all accurate" but "completely explainable" AI (see Fig. 5a and SI Notes, Materials Study 3B for additional instructions). After reading through the instructions that provided definitions of AI, explainability, and accuracy and after successfully passing comprehension checks, participants were presented with the five AI applications from Studies 2 and 3A. Because Study 2 indicated stakes and scarcity as factors shaping participants' valuation of interpretability, we again included four versions of each application, varying by stakes and scarcity. For each application-version, participants used the described tradeoff-slider to indicate whether they would prefer a more interpretable but less accurate, or a less interpretable but more accurate AI. As we were interested in people's attitudes or a priori preferences, we continued using the scenario format of our first studies, where we did not specify the outcome of the machine-made decisions. Previous work from psychology suggests people might perceive AI-accuracy as a proxy, or at least precondition, for ensuring favourable outcomes[28,31], which would suggest an overall preference for accuracy over interpretability.

We coded participants' responses such that positive values represented a preference for interpretability over accuracy and negative values indicated a preference for accuracy over interpretability. Our data revealed an overall preference for accuracy over interpretability, signified by a mean rating of $M = -0.36$ that differed significantly from the indifference point of 0, $t(2,239) = -12.21$, $p < 0.001$, 95% CI [−0.41, −0.30].

Next, we ran a linear mixed effects model predicting participants' tradeoff preferences, with stakes, scarcity, and their interaction entered as fixed effects while we entered participant and application as random intercept effects. Type II Wald $\chi^2$ tests indicated significant main effects for stakes ($\chi^2(1) = 52.91$, $p < 0.001$) and scarcity ($\chi^2(1) = 24.42$, $p < 0.001$) on tradeoff preferences, which were not qualified by an interaction, $\chi^2(1) = 1.13$, $p = 0.288$ (Fig. 5b). Overall, participants preferred accuracy over interpretability, and this preference was amplified by the same conditions that impacted preferences for interpretability in Study 2. That is, participants' preferences for accuracy over interpretability were more pronounced for high relative to low stakes cases ($b = -0.42$, $p < 0.001$, $d = 0.12$, 95% CI [0.05, 0.20]) and for cases involving high relative to low scarcity,

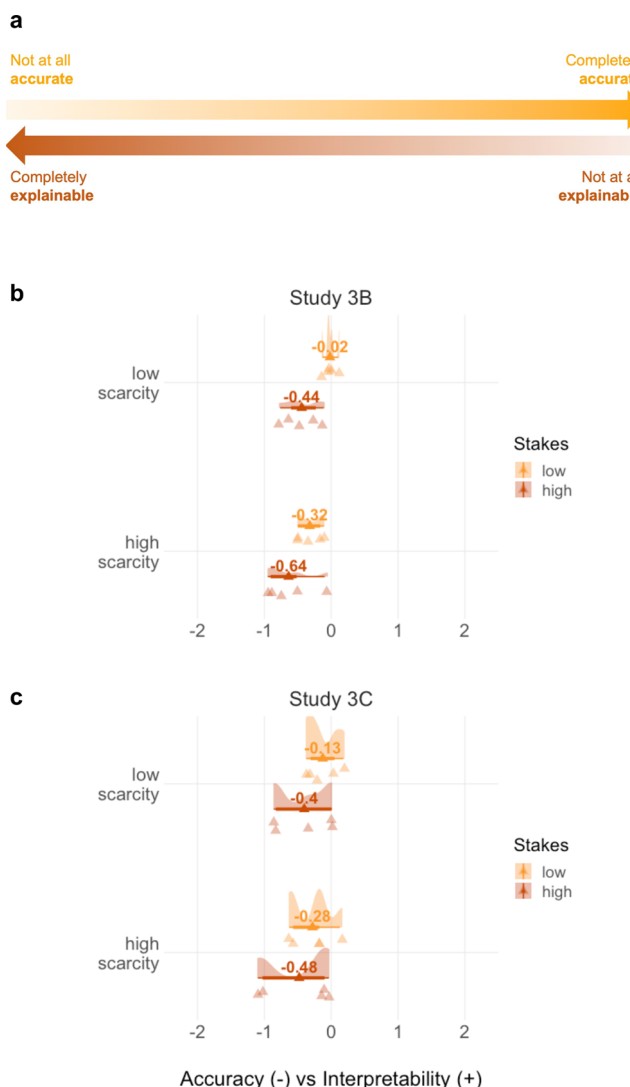

**Fig. 5 | Dependent variable and results for Studies 3B and 3C. a** Dependent variable on which participants were asked to move the slider to a position representing their preference for the interpretability - accuracy tradeoff. The order of attributes and hence the direction of the slider was counter-balanced across participants. **b** Tradeoff-preferences from Study 3B (*N* = 112; within-subjects design), aggregating across all five applications. **c** Tradeoff-preferences from Study 3C (*N* = 1344; between-subjects design), aggregating across all five applications.

*b* = −0.30, *p* < 0.001, *d* = 0.09, 95% CI [0.01, 0.17]. These effects were robust to controlling for AI- and task-related covariates, in particular the ordering of accuracy and interpretability across instructions and the response-variable, pre- and post-task support for AI, and computer science knowledge (see SI Results, Study 3B). Main effects for stakes and scarcity also remained significant when we added further explanatory candidates, such as decision-reversibility or personal affectedness, to the model (see SI Results, Study 3B).

The results of Study 3B suggest that people prioritize AI accuracy over interpretability when the two trade off against one another. Moreover, participants appear to be more inclined to sacrifice interpretability for accuracy under the same conditions under which they value interpretability most when considered on its own (i.e., high stakes and high scarcity). In Study 3C, we sought to replicate these findings in a US sample nationally representative for age, race, and gender, and using a between-subjects design that reduces the salience of differences in (low versus high) stakes and scarcity.

## Study 3C–replicating effects of stakes and scarcity on interpretability-accuracy tradeoffs

Participants in Study 3B were presented with four different versions of each AI application, which might have increased the salience of variation in stakes and scarcity. This, in turn, might have enhanced participants' sensitivity to variations in stakes and scarcity[33,34]. Thus, in Study 3C, we sought to test the robustness of our findings using a between-subjects design in which each participant was presented with only one combination of stakes and scarcity. Our sample (final *N* = 1344; recruited from Prolific) was representative of the US population in terms of its gender by age by race composition. Participants were randomly allocated to one of four between-subjects conditions (low stakes, low scarcity; low stakes, high scarcity; high stakes, low scarcity; high stakes, high scarcity) and presented with each of the five applications from Studies 2 and 3A. Similar to our previous studies, participants were given a general description of a given application that mentioned how stakes and scarcity could be low or high before specifying the exact combination according to the between-subjects manipulation. For each application, participants stated their preferences on the slider measure from Study 3B, where one end represented a "completely accurate" but "not at all explainable" AI, whereas the other end represented a "not at all accurate" but "completely explainable" AI. All other instructions and comprehension checks were the same as in Study 3B. The pre-registered procedure, hypotheses, and analysis plan are available at the Open Science Framework[32].

Again, we coded participants' responses such that positive values represented a preference for interpretability over accuracy and negative values indicated a preference for accuracy over interpretability. In line with our findings from Study 3B, we observed an overall preference for accuracy over interpretability, signified by a negative average of *M* = −0.32 that differed significantly from the indifference point, *t*(6,719) = −19.00, *p* < 0.001, 95% CI [−0.36, −0.29].

Next, we ran a linear mixed effects model predicting participants' tradeoff preferences, with stakes, scarcity, and their interaction entered as fixed effects while we entered participant and application as random intercept effects. Type II Wald chi-square tests indicated significant main effects for stakes ($\chi^2(1) = 34.18$, *p* < 0.001) and scarcity ($\chi^2(1) = 7.84$, *p* = 0.005) on tradeoff preferences, which were not qualified by an interaction, $\chi^2(1) = 0.93$, *p* = 0.336 (Fig. 5c). The main effects of stakes (*b* = −0.28, *p* < 0.001, *d* = 0.06, 95% CI [0.01, 0.11]) and scarcity (*b* = −0.15, *p* = 0.008, *d* = 0.03, 95% CI [−0.02, 0.08]) on tradeoff preferences thus replicated in the between-subjects design that minimised salience of variation in the two attributes. And again, main effects for stakes and scarcity remained significant when we added further explanatory candidates, such as decision-reversibility or personal affectedness, to the model (see SI Results, Study 3C). However, effect sizes relative to Study 3B were extremely small. This suggests that people's sensitivity to stakes and scarcity is dependent on the salience of variation in the two attributes, which was higher in the within-subjects design than the between-subjects design. Indeed, as we report in Study 3D in the SI, when we ran an additional experiment that reduced the salience of variation of the two attributes to a minimum, by not even mentioning their range, only the main effect for stakes remained significant (*p* < 0.001) whereas the effect for scarcity was no longer significant (*p* = 0.136).

Over time, as the use of AI spreads ever more widely, people will be increasingly likely to encounter variations of stakes and scarcity within and across AI applications in the real-world. This will arguably enhance people's sensitivity to stakes and scarcity present in a given AI application and foster the formation of more systematic and stable preferences over accuracy and interpretability in AI[34]. But already at this point, where most people's awareness and experience of interacting with AI remains scattered, our findings suggest that people's attitudes are sensitive to variations in stakes and scarcity both across

applications (Studies 1A–1C), as well as within applications (Studies 2, 3B, 3C).

## Discussion

In recent years, academics, policymakers, and developers have debated whether interpretability is a fundamental prerequisite for trust in AI systems. However, it remains unknown whether non-experts–who may ultimately comprise a significant portion of end-users for AI applications–actually care about AI interpretability, and if so, under what conditions. Here, we characterise public attitudes towards interpretability in AI across seven studies. Our data demonstrates that people consider interpretability in AI to be important. Even though these positive attitudes generalise across a host of AI applications and show systematic patterns of variation, they also seem to be capricious. While people valued interpretability as similarly important for AI systems that directly implemented decisions and AI systems recommending a course of action to a human (Study 1A), they valued interpretability more for applications involving higher (relative to lower) stakes and for applications determining access to scarce (relative to abundant) resources (Studies 1A-C, Study 2). And while participants valued AI interpretability across all levels of AI accuracy when considering the two attributes independently (Study 3A), they sacrificed interpretability for accuracy when these two attributes traded off against one another (Studies 3B–C). Furthermore, participants favoured accuracy over interpretability under the same conditions that drove importance ratings of interpretability in the first place: when stakes are high and resources are scarce.

Our findings highlight that high-stakes applications, such as medical diagnosis, will generally be met with enhanced requirements towards AI interpretability. Notably, this sensitivity to stakes parallels magnitude-sensitivity as a foundational process in the cognitive appraisal of outcomes[35,36]. The impact of stakes on attitudes towards interpretability were apparent not only in our experiments that manipulated stakes within a given AI-application, but also in absolute and relative levels of participants' valuation of interpretability across applications–take, for instance, 'hurricane first aid' and 'vaccine allocation' outperforming 'hiring decisions', 'insurance pricing', and 'standby seat prioritizing'. Conceivably, this ordering would also emerge if we ranked the applications according to the scope of auditing- and control-measures imposed on human executives, reflecting interpretability's essential capacity of verifying appropriate and fair decision processes[7,26,37,38].

Fairness concerns are also salient in 'gatekeeping' settings where decision-makers determine access to scarce resources[27–29]. Accordingly, we found that the importance of interpretability was higher when AI applications allocated resources under conditions of scarcity. These findings build on past work showing that people demand more explanation for decisions involving resource allocation in order to ensure that the allocation process was fair[11,38,39], demonstrating that such principles also operate in the context of AI applications and substantiating calls for interpretability as a safeguard for ethical and fair AI systems. Enhanced valuation of interpretability in such settings seems all the more justified and important in view of recent anecdotal evidence that (apparent) lack of interpretability may provide human agents in charge of overseeing outcomes produced by AI systems with the opportunity to obscure personal responsibility: when allocation decisions for vaccines against Covid-19 went awry, prioritising administrators before frontline healthcare workers, responsible officials blamed a "very complex algorithm" for the undesirable outcomes[40]. The fact that this algorithm turned out to be a relatively simple and hand-coded rule-based formula[41] highlights the danger that humans in charge may purport lack of interpretability in AI even when this is not the case.

In practice, AI interpretability and AI accuracy come often–but not necessarily[13]–as a tradeoff. When we explored participants' attitudes towards interpretability without imposing such a tradeoff, we found that most participants rated interpretability as invariably important across all levels of AI accuracy, indicating they value interpretability in AI in its own right. In contrast, when we confronted participants with a tradeoff between AI interpretability and AI accuracy, they sacrificed interpretability for accuracy, and were more inclined to do so for high-stakes applications and those involving the allocation of scarce resources. Prioritizing accuracy over interpretability by seeking "answers first, explanations later" accrues what the legal scholar Jonathan Zittrain has described as "intellectual debt"[42]: answers gained at the expense of understanding. Intellectual debt is risky because lacking understanding of how something works can produce negative unintended consequences in complex systems. For instance, if a drug is effective but the underlying mechanism is unknown, prescribing that drug can lead to dangerous side-effects if administered in combination with other drugs. Likewise, accruing intellectual debt in AI systems becomes riskier in settings where multiple AI applications will interact: consider a medical system where AI diagnosis applications are used in combination with AI applications that decide who gets access to scarce medical treatments. Our studies imply that even though participants value interpretability in its own right, they endorse the accrual of intellectual debt when interpretability trades off with accuracy. In fact, they were most inclined to sacrifice interpretability for accuracy under conditions of high stakes and high scarcity–those conditions where negative unintended consequences are likely to produce the most damage.

As much as the present work offers a glimpse into public attitudes towards interpretability, it also highlights the need for deeper insights. Where scholars grapple with conceptual and practical controversies about interpretable AI[13,43], non-experts arguably have an even harder time to understand the concepts at hand. As much as this may justify our relatively liberal and simplified definitions of interpretability and accuracy, it also constitutes a limitation of our work. For instance, the prominence of "mistakes" in our definition of accuracy ("the more accurate an AI, the fewer mistakes it makes when performing decision") might have inflated people's valuation of accuracy relative to interpretability in Studies 3B and 3C. Furthermore, the conclusions drawn from the present work are limited to participants from the US and the UK. Exploring attitudes towards interpretable AI among other populations is a promising and important topic for future work, especially in light of recent work suggesting that expectations towards machine-made decisions can vary substantially across countries and cultures[44] and amidst reports about the potential of disproportionately harmful impacts of AI on the lives of low-income populations[45].

As the technical implementation of different degrees of interpretability in AI develops, policy-makers and users alike might update their a priori attitudes towards interpretability in AI. It will then become possible and important for future research to characterise how the findings described in the present work depend on stakeholder perspectives (e.g., policy-makers versus users) and how they evolve over time, to explore how attitudes towards interpretability translate into manifest choices, and how they relate to attitudes towards explanations of human decisions[46]. For instance, a scenario conceivable in the near future might be that healthcare providers will offer patients a choice between a version of a medical algorithm that vastly outperforms human doctors in medical diagnosing but that is not at all interpretable[3], or a version that performs slightly better than human doctors and that is interpretable. It will be important to characterise people's revealed preferences in such settings, which will also allow to explore whether, and if so how, valuations of interpretability differ between a priori appraisals, where outcomes are unknown and on which we focused in the present work, as opposed to a posteriori appraisals, where outcomes are known. Valuations might also depend on stakeholder perspectives (e.g., human patient versus human

doctor), even though findings from an explorative experiment that we report in the Supplementary Information (Supplementary Results, Study 4) indicate no differences in terms of stated a priori attitudes towards interpretability for perspectives of an affected patient versus a responsible agent.

Given the lack of empirical work on people's valuation of interpretability in AI, our contribution started by considering two variables–stakes and scarcity–that have emerged as important from existing literature on appraising the "good-" and "fairness" of human-made decisions. But the nature and scale of machine-made decisions warrants for considering further features of decision situations and their role in people's valuation of interpretability[46,47]. For instance, in view of recent work indicating that perceptions about required human expertise in a given decision-context affect people's willingness to rely on algorithmic advice in that context[48], we explored 'required human expertise' as an additional explanatory variable and find tentative evidence that perceiving high expertise-requirements might sway tradeoff-preferences away from prioritising interpretability, towards prioritising accuracy (see SI Results, Studies 3B, C). Similarly, an important avenue for future research will be to further characterise the relationship between different features of AI systems. Our results from Studies 3A–C indicate that while people have dissociable preferences for individual AI features, complex patterns and interactions emerge when technological constraints, such as interpretability-accuracy tradeoffs, are considered. While the present work reflects the focus on AI interpretability and accuracy as hallmarks of desirable AI, we hope that it generates further research into other AI properties such as transparency or usability.

Answering the most basic of all questions–"why?"–is a fundamental human need[49]. Our findings show that this quest for explanation translates into positive attitudes towards interpretability in AI systems, which are pronounced most strongly in applications involving high stakes and scarce resources. Nevertheless, when interpretability trades off against accuracy, people are willing to accrue intellectual debt in these same settings, seeking accurate solutions at the expense of understanding. These findings highlight the importance of further characterising and respecting human attitudes towards AI interpretability, particularly when designing and/or regulating complex technical systems with high stakes for human welfare.

## Methods

All studies were approved by the Yale University Human Participants Committee (approval number: HSC 2000022385) and participants in each study gave their informed consent beforehand. Participants for all studies were recruited via MTurk or Prolific. MTurk and Prolific provide more diverse participant pools than university students[50–52], including representative samples of certain populations, which we recruited as we were interested in the general public's views on interpretability in AI. All participants were paid in line with minimum wages for the US ($7.25 hourly rate for Studies 1A, B; 2–4) and for the UK (£8.72 hourly rate for Study 1C). The exact instructions for all studies are deposited as Qualtrics questionnaires at the Open Science Framework under https://doi.org/10.17605/OSF.IO/DQ4VC.

### Study 1A
**Participants.** We recruited 200 participants from the US via MTurk (data collected 20/04/2019). One duplicate response and 29 participants who failed a comprehension check on their second attempt were excluded from our analyses, leaving a final sample of $N = 170$. The final sample included 107 males, 60 females, and 3 "other" with an average age of 36.94 (SD = 11.83, SE = 0.91). In all, 20 of the participants' highest education was high school; 40, some college; 16, a 2-year degree; 75, a 4-year degree; and 19, a postgrad or another professional degree. The mean income bracket was between $35,001 and $50,000. Most participants (74) had no formal education in computer science; 35 had

some programming experience; 45 took a college-level course; 13 held an undergraduate degree; and 3 held a graduate degree in computer science.

**Procedure.** Participants learned that we were interested in their attitudes towards AI, which we defined as follows[53]: "Artificial Intelligence (AI) refers to computer systems that make predictions, recommendations, or decisions by learning from existing datapoints. This computerised process is automated and occurs without explicit human instructions". They proceeded to questions about their intuitions on "the extent to which people [with/without] training in computer science can explain how an AI reaches certain predictions, recommendations, or decisions in certain cases" (5-point rating scale from "cannot explain at all" to "can explain fully") and their general support for AI (5-point rating scale from "strongly oppose" to "strongly support"). Next, we provided them with a precise definition of explainability reading "By explainable we mean that an AI's [decision/recommendation] can be explained in non-technical terms. In other words, it is possible to know and to understand how an AI arrives at its [decision/recommendation]". This was followed by a simple comprehension that all participants were ultimately allowed to pass, though we excluded those who failed on two attempts from our analyses (probe: "What is meant by 'explainable AI'?"; correct response: "That an AI's [decision/recommendation] can be explained in non-technical terms"). Participants then learned that they would go through a series of AI applications for each of which they would answer "how important is it that the AI in this application is explainable, even if it performs accurately" on a dichotomous 5-point rating scale from "not at all important" to "extremely important". Each participant saw either the recommend or decide version of the application-descriptions. We had a total of 27 decide applications and 25 recommend applications (see SI Notes, Materials Study 1A for details) and each participant saw a random subset of 20 out of these, presented on subsequent pages with a reminder about explainability's definition displayed at the bottom ("Reminder: Explainable means that the AI's recommendation can be explained in non-technical terms"). After providing their ratings for the applications, participants indicated how important they considered the respective motives "explain to justify/to verify/to improve/to discover"[53] (see SI Results, Study 1A). We then probed them again on their support for AI and asked how likely they considered it that their occupation would be replaced by AI at some point, and whether they had any computer science knowledge. The survey concluded with standard demographics (gender, age, income, education).

### Study 1B
**Participants.** We recruited 293 US participants via Academic Prolific, using the platform's feature for collecting representative samples that match census data in terms of age by sex by ethnic group proportions (data collected 14/01/2021). Of those, in line with the pre-registration, we excluded 33 participants who failed comprehension checks on both attempts, or who failed all attention checks. The final sample of $N = 258$ included 125 males, 131 females, 1 nonbinary person, and 1 who chose "prefer not to say" with an average age of 45.53 (SD = 16.41, SE = 1.02). 2 of the participants' highest education was less than high school; 22, high school; 49, some college; 17, a 2-year degree; 112, a 4-year degree; and 56, a postgrad or another professional degree. The mean income bracket was between $35,001 and $50,000. Most participants (129) had no formal education in computer science; 42 had some programming experience; 65 took a college-level course; 15 held an undergraduate degree; and 7 held a graduate degree in computer science.

**Procedure.** Participants received the same definitions and instructions as in Study 1A. However, across instructions, comprehension checks, and dependent variable questions, we replaced the word "explainable"

with "understandable" as a robustness check. Thus, as participants went through the AI applications, they answered the question "In this case, how important is it that the AI is understandable?" for each one. In order to allow for more fine-grained responses, we changed the answer format from a dichotomous measure to a continuous slider with five tick marks and the same three labels as in Study 1A. To comply with the data collection policy of Prolific, participants were not screened out during the survey when they met the pre-registered exclusion criteria (i.e., failing comprehension checks on both attempts and/or all attention checks). We added a comprehension check to verify that participants understood they should assume the AI to perform accurately (probe: "What should you do in the following task?"; correct response: "Consider how important it is that an AI is understandable, even if it performs accurately") and an attention check that was not related to the main task. Analogously, we extended the reminder displayed below the answer variable so that it read "'understandable' means that the AI's decision can be explained in non-technical terms. Please consider how important it is that the AI is understandable, even if it performs accurately". We collected the same control and demographic variables as in Study 1A.

## Study 1C

**Participants.** We recruited 298 UK participants via Academic Prolific, using the platform's feature for collecting representative samples that match census data in terms of age by sex by ethnic group proportions (data collected 14/01/2021). Of those, in line with the pre-registration, we excluded 52 participants who failed comprehension checks on both attempts, or who failed all attention checks. The resulting final sample of $N = 246$ included 118 males and 128 females with an average age of 45.57 (SD = 15.94, SE = 1.02). 28 of the participants' highest education was less than high school; 57, high school; 102, some college; and 59, a two-year degree. The mean income bracket was between $15,001 and $25,000. Most participants (162) had no formal education in computer science; 50 had some programming experience; 18 took a college-level course; 9 held an undergraduate degree; and 7 held a graduate degree in computer science.

**Procedure.** We applied the exact same instructions and procedures as for Study 1B.

## Study 2

**Participants.** We recruited 120 US participants via MTurk (data collected 24/10/2019). 36 participants failed comprehension checks on two attempts and were excluded from all analyses, resulting in a final sample of $N = 84$. The final sample included 46 males, 36 females, 1 "other", and 1 "prefer not to say" with an average age of 37.38 (SD = 11.76, SE = 1.28). In all, 13 of the participants' highest education was high school; 25, some college; 7, a 2-year degree; 31, a 4-year degree; and 8, a postgrad or another professional degree. The mean income bracket was between $25,001 and $35,000. In total, 5 of the participants had no formal education in computer science; 42 had some programming experience; 14 took a college-level course; 18 held an undergraduate degree; and 5 held a graduate degree in computer science.

**Procedure.** Participants were presented with the same definitions of AI and explainability as in Study 1. We added two comprehension check questions to ensure they understood the definition of explainability and that their task was to consider the importance of explainable AI assuming it would perform accurately. In order to allow more fine-grained responses, we slightly changed answer format to a continuous slider with five tick marks and the same three labels as before ("not at all important", "moderately important", "extremely important"). We fully randomized the order of applications across participants, as well as the order of versions within each application (high/low stakes and

scarcity). We also extended the reminder displayed below the answer variable so that it read "Explainable means that the AI's decision can be explained in non-technical terms. Please consider how important it is that the AI is explainable, even if it performs accurately". We collected the same additional variables as in Studies 1 and 2, plus a question asking whether they had heard about bias in AI, and if so, whether they thought their answers had been influenced by this.

## Study 3A

**Participants.** We recruited 302 US participants via Academic Prolific (data collected 26-28/05/2021), using the platform's feature for collecting representative samples that match census data in terms of age by sex by ethnic group proportions. In line with our pre-registration, we excluded 41 participants who failed a comprehension check on two attempts, leaving a final sample of $N = 261$. The final sample included 124 males, 130 females, 4 nonbinary, and 1 "prefer not to say" with an average age of 44.88 (SD = 15.80, SE = 0.98). In all, 24 of the participants' highest education was high school; 42 some college; 30, a two-year degree; 109, a four-year degree; and 56, a postgrad or another professional degree. The mean income bracket was between $35,001 and $50,000. Most participants (134) had no formal education in computer science; 47 had some programming experience; 60 took a college-level course; 15 held an undergraduate degree; and 5 held a graduate degree in computer science.

**Procedure.** Participants were presented with the five vignettes that we used for Study 2. However, this time we only used the introductory part that outlined the general scenario (e.g., "It is flu season. An AI decides whether or not a citizen will get a vaccine"). For each vignette, participants were asked rate "How important is it that the given AI model is explainable" on a slider ranging from "not at all important" to "extremely important". Importantly there always were four sliders, representing four different AI models that differed in accuracy. Our instructions sought to highlight the difference between those models by explaining that "Each slider represents a different AI model. The first slider represents an AI model performing at 90% accuracy; the second slider represents a different AI model that performs at 80% accuracy, and so forth until the fourth slider, which represents again a different AI model that performs at 60% accuracy". We counterbalanced the order or AI models across participants. We also provided participants with some exemplary importance ratings, such as "For some applications, you might think that an AI model being explainable is equally important for any level of accuracy. In these cases, you would move all sliders to the same level" or "For other applications, you might think that the importance of an AI model being explainable depends on how accurately it performs. In these cases, you would move the sliders representing AI models differing in accuracy to different levels" (all instructions are detailed in the pre-registration available at the Open Science Framework under https://doi.org/10.17605/OSF.IO/DQ4VC).

## Study 3B

**Participants.** We recruited 120 US participants via MTurk (data collected 15-16/11/2019). To reduce the dropout from failed comprehension checks that we had witnessed in Study 2, we only let those participants proceed to the full survey, who successfully passed comprehension checks. Participants who failed to do so on two attempts were paid a compensation fee for their time until that point and were replaced. This left us with a final sample of $N = 112$ participants after excluding five duplicate submissions. The final sample included 77 males, 34 females, and 1 "prefer not to say" with an average age of 36.45 (SD = 10.30, SE = 0.97). In total, 14 of the participants' highest education was high school; 22, some college; 11, a 2-year degree; 50, a 4-year degree; and 15, a postgrad or another professional degree. The mean income bracket was between $35,001 and $50,000. Most participants (45) had no formal education in computer science;

21 had some programming experience; 29 took a college-level course; 7 held an undergraduate degree; and 10 held a graduate degree in computer science.

**Procedure.** Participants first read definitions of AI and AI explainability. In order to align our instructions across studies and to reflect the continuous answer variables used in this study, we slightly amended the definition of AI explainability such that it read "By 'explainable' we mean that it is possible to understand how an AI arrives at its decision. In other words, the more explainable an AI, the more one can understand about how it makes its decisions. The less explainable an AI, the less one can understand about how it makes its decisions". Additionally, participants were presented with a definition of 'accuracy' reading as follows: "By 'accurate' we mean that an AI's decisions are correct. In other words, the more accurate an AI, the fewer mistakes it makes when performing decisions. The less accurate an AI, the more mistakes it makes when performing decisions. The instructions also highlighted that, ideally, one would want an AI that is both completely accurate and completely explainable, but that, in practice, a more accurate AI is often less explainable; vice versa, a more explainable AI is often less accurate. Correspondingly, we amended our comprehension checks and added two additional questions ensuring that participants had understood the definition of accuracy and the tradeoff response-format. Participants then were presented with the five applications and their four respective versions in randomized order. On each round, they used the tradeoff-slider to indicate their preferences over interpretability and accuracy by moving the slider's button from the middle position to either end. The ordering of the accuracy and explainability labels, and correspondingly the valence of the slider measure's ends, was counter-balanced across participants. Participants were reminded that "Accuracy depends on correctly determining which standby passengers need to board urgently. Explainability depends on understanding the criteria used to determine which standby passenger need to board urgently", with the reminder order mirroring the order of the two attributes encountered in previous instructions and the slider set-up assigned to a given participant. In order explore additional explanatory factors in people's preferences for accurate and interpretable AI that have been identified as relevant or desirable in ethical frameworks on AI governance as well as empirical research on preferences for the use of AI, we added randomly ordered questions probing participants' perception of (i) the reversibility for the AI's decisions[54], (ii) the level of expertise required for a human to perform the AI's decision[48,55], (iii) the likelihood of being personally affected[53], and (iv) the number of people affected[53]. Participants answered these measures separately for each of the applications from the main task. Otherwise, we collected the same additional variables as in Study 2.

## Study 3C
**Participants.** In line with our pre-registration, we recruited 1501 US participants via Prolific (data collected 25-27/08/2021), using the platform's feature for collecting representative samples that match census data in terms of age by sex by ethnic group proportions. After excluding 27 incomplete and duplicate responses, as well as 127 participants who failed a comprehension check on two attempts in line with the pre-registration, we obtained a final sample of $N = 1344$. The final sample included 720 males, 605 females, 1 nonbinary, and 16 "prefer not to say" with an average age of 39.26 (SD = 14.90, SE = 0.41). In all, 8 of the participants' highest education was less than high school; 103, high school; 279, some college; 106, a 2-year degree; 422, a 4-year degree; and 423, a postgrad or another professional degree. The mean income bracket was between $35,001 and $50,000. Most participants (610) had no formal education in computer science; 211 had some programming experience; 367 took a college-level course; 44 held an undergraduate degree; and 111 held a graduate degree in computer science.

**Procedure.** Participants generally followed the same instructions and procedures as in Study 3B. However, this time *stakes* and *scarcity* varied only between participants: Each participant was randomly allocated to one of the four conditions (low stakes, low scarcity; low stakes, high scarcity; high stakes, low scarcity; high stakes, high scarcity). They remained in the same condition while going through the five different vignettes (allocating vaccines, prioritizing first responders, reviewing insurance claims, making hiring decisions, allocating standby passengers) that were randomised in order with the only exception that allocating vaccines always was the first vignette. We slightly altered the vignettes from Study 3B such that they mentioned the possible range stakes and scarcity (e.g., mild or deadly flu; abundant or very limited vaccine supply; see SI Methods, Materials Study 3C). Otherwise, all instructions, materials, and procedures followed Study 3B.

## Reporting summary
Further information on research design is available in the Nature Research Reporting Summary linked to this article.

## Data availability
All original data (de-identified) are available at the Open Science Framework (https://doi.org/10.17605/OSF.IO/DQ4VC).

## Code availability
All analysis code, including the code for data preparation, is available on the Open Science Framework (https://doi.org/10.17605/OSF.IO/DQ4VC).

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

## Acknowledgements
We thank Nisheeth K. Vishnoi and members of the Crockett Lab for helpful feedback. M.J.C. was supported by a grant from the John Templeton Foundation (#61495). This publication was supported by the Princeton University Library Open Access Fund.

## Author contributions
A.-M.N., L.L., E.C., and M.J.C. designed research; A.-M.N. and L.L. performed research; A.-M.N. and L.L. analysed data with input from M.J.C.; and A.-M.N., E.C., and M.J.C. wrote the paper.

## Competing interests

The authors declare no competing interests.
