## [Peer Review File · Nature Communications]

Public attitudes value interpretability but prioritize accuracy in artificial intelligenceREVIEWER COMMENTS

Reviewer #1 (Remarks to the Author):

This study considers a very interesting and timely question; human preferences for interpretability in AI-based systems and how that compares to preferences on accuracy. In so doing, the authors present four well designed studies, and provide an appropriate discussion. The study contains various interesting findings that would be very informative and impactful. However, I think there are some crucial issues to address:

1. The main message is not very clear. The paper reads as a collection of interesting results about preferences for interpretability in relation to different factors. The most interesting part for me is Study 4, specifically the comparison and interplay between interpretability and accuracy. But even this relationship is not investigated in sufficient depth. This study tells us that people prefer accuracy over interpretability, both when traded off and when taken independently (and regardless of stakes and scarcity). This misses an important and plausible point; preference for interpretability is almost irrelevant for an inaccurate system. An AI that has low accuracy (beyond some threshold) is undesirable regardless of how interpretable it is. It is therefore crucial to investigate the tradeoff in a different way, probably starting from an acceptable-accuracy threshold point rather than from the 'not at all accurate' point. Moreover, demand for interpretability could be dependent on accuracy (but not vice versa). In this case, the question could better be asked as: How strongly would people demand interpretability from an AI that has 100% accuracy? What about an AI with 90% or 80% accuracy? I imagine that demand for interpretability as a function of model accuracy will take a monotonically decreasing shape (after some acceptable-accuracy threshold). But it is an important and an open question of how this would really look like (smooth continuous decrease vs. one big drop), and whether it changes based on the application (medical diagnosing vs. parole reviewing).

2. Participants are recruited from MTurk, which does not provide nationally representative samples. The authors do not discuss this point in the manuscript. How do we know that these results hold for the whole population? Discussing this in the paper is a bare minimum, but I don't think it is sufficient for a publication in this journal. The authors need to show that the results (at least the ones in Study 4) would generalize to the population residing in the US, either by replicating the study through platforms that provide representative samples, or (less preferably) doing some post-stratification or sensitivity analysis.

3. The findings of this study about preferences is limited to 'stated preferences'. I may have missed it, but I don't think this point is mentioned in the manuscript. So, is the goal of the paper capturing stated preferences? Or does the paper rely on the assumption that stated preferences are not very different from revealed preferences in this context? I understand it's probably not easy (yet not impossible) to devise a study to capture revealed preferences in this case, but the authors need to discuss this point, and either show empirical evidence for this assumption or argue why they think it is sensible in this case. Only acknowledging this as a limitation is not enough.

4. This study is limited to American participants (or participants residing in the United States), a point acknowledged in the Discussion (but only there). That the participants are Americans or live in the US should be mentioned at least in the Methods section and in the main manuscript at the beginning of each study when describing participants and N.

5. The effect of scarcity and stakes is studied in a within-subject design in Study 2-4. It is never varied between subjects. The authors need to at least show that a carryover effect is unlikely here (which they can do given their order randomization), or (even better) re-run the study in a between-subject design.

6. How were the annotations of scarcity and stakes levels in Study 1 done? Was that done by one of

the authors or was it done by participants? It seems like it was done by the authors, which would be a problem. I think neither of the two factors is perfectly objective, and they could be open for interpretation. For example, medical diagnosing could be thought of as a low stakes, and automated grading is not likely to be understood as a scarce commodity case (unless you're grading on a curve). I know that the authors used this as an exploration point which was properly designed for in Study 2, but I still think that the correlational evidence in Study 1 add another type of evidence (more useful for prediction than inference) about tasks that are innately high/low stakes and (not) scarce. If this was not done by the participants, I'd suggest asking 1-3 people who are not familiar with the manuscript findings to do the annotation.

Reviewer #2 (Remarks to the Author):

This paper presents four studies investigating public demand for interpretable AI. In these studies, participants from an online panel rated how important they thought it was for AIs to be interpretable in various applications, or their preferred tradeoff between AI interpretability and accuracy. The results show that people generally think it is important for AIs to be interpretable, especially when the stakes of the decision are high and when the decision involves a scarce resource. However, when contrasted to accuracy, people showed a greater preference for accuracy over interpretability for decisions involving high-stakes and scarce resources. The perspective participants were asked to take – that of a person affected by the decision or of a person responsible for managing the AIs decision – did not affect people's preference for accuracy over interpretability.

To the best of my knowledge, the paper is novel. Indeed, as the authors mention, questions of AI interpretability have been discussed, but there are no systematic investigations of public demand of interpretable AIs.

A systematic investigation of public demand for interpretable AIs could indeed be of interest to many in the field of human-robot interaction. Although the paper in its current form does not significantly advance theory in the field, a systematic test of whether laypeople show the same considerations engineers and ethicists discuss is important.

Several aspects of the paper can be strengthened. I expand on several suggestions below. Some of these suggestions are more difficult to implement than others. These should be viewed as suggestions, and not all of them should be seen as necessary.

The paper could benefit from more diversity in the measurement of "demand for interpretable AI". Currently, two ways are used to measure "demand". 1) self-reported "importance" (studies 1-2); 2) self-reported preference for the interpretability-accuracy tradeoff (studies 3-4). I am not sure that either of these measures necessarily reflects "demand", at least not in the economic sense. "Importance" or "preference" might be more accurate terms. The authors can strengthen the paper by expanding its measurement. For example, by examining how people rely on various AIs on the interpretable-accurate tradeoff for a joint task. Another possibility is to explore the market response to various AI systems that vary in their interpretability and accuracy. Are certain systems more popular than others? These are just a few examples, I am sure there are many more. I do think, however, that the current measurement needs strengthening.

As the paper sets out the measure "public demand", a more comprehensive sample (or samples) could also strengthen the argument about "public". Relying on one online platform and using relatively small samples limits generalizability. This is especially important since the goal of the paper is to investigate "public demand". Although this limitation is mentioned in the discussion, using larger samples, collected on multiple platforms, could strengthen the paper and the argument it makes. This will also allow exploring individual and cultural differences.

One reason why the idea of AI interpretability became popular, is that it is seen as a way to safeguard from discrimination (Gilpin et al., 2019). Interpretability is sometimes thought of as a way to create responsible, accountable and fair AI systems. Of course, one paper cannot examine all possible factors. However, another way to further strengthen the paper is to explore an interpretability-fairness tradeoff.

Another way to further strengthen the paper is by uncovering the psychological mechanisms underlying the demand for interpretability. Why do people want interpretability? Why do stakes and scarcity affect the importance of interpretability? Is there a psychological difference between the importance of understanding a decision by an AI and a decision by a human agent? Are there cases where people insist on interpretability even at the cost of reduced accuracy? Addressing any of these questions can enhance our understanding of the phenomenon and the contribution of the paper to theory.

All of the experiments used a within-subject design. Such designs can indeed increase the salience of some attributions (Hsee et al., 1999). The paper could be strengthened by examining the pattern of preferences also in a between-subject design. Manipulating AIs within-subjects can be easily justified, as people might need to choose between two AIs, one more accurate and one more interpretable. Regardless, the paper should justify the choice of experimental design and its advantages and limitations.

A few minor comments:

Interpreting the null effect for perspective in Study 4 should state more clearly the limitations of not finding a difference with NHST (Null Hypothesis Significance Testing). The authors can strengthen the paper by reporting the statistical power of their experiment or, preferably, by using a Bayesian approach.

Including the full study materials in the supplemental materials would be very helpful. This should include the full studies and the exact text of all collected variables. Reading the paper, it was sometimes unclear exactly how the measurement was done, and the methods section or the SI did not provide enough detail. For the sake of transparency, I would suggest clarifying whether other variables were collected as well. As the studies were not pre-registered, such a clarification would be helpful, and make the studies easier to reproduce.

Similarly, describing the statistical analysis in more detail could be helpful. Some analyses (such as Study 3, in lines 243-248) have more detail than others. Adding the full details of all reported analyses could be helpful. In some cases, adding justifications for the statistical approach could be beneficial as well. For example, explaining why sometimes an ordinal regression was used and sometimes a linear regression.

Finally, pre-registration and discussion of statistical power are becoming standard in experimental psychology. I understand that it might be a little late for pre-registration. However, I recommend pre-registering any additional study that might be conducted.

Gilpin, L. H., Bau, D., Yuan, B. Z., Bajwa, A., Specter, M., & Kagal, L. (2019). Explaining explanations: An overview of interpretability of machine learning. *Proceedings - 2018 IEEE 5th International Conference on Data Science and Advanced Analytics, DSAA 2018*, 80–89.

<https://doi.org/10.1109/DSAA.2018.00018>

Hsee, C. K., Loewenstein, G. F., Blount, S., & Bazerman, M. H. (1999). Preference reversals between joint and separate evaluations of options: A review and theoretical analysis. *Psychological Bulletin*, 125(5), 576–590. <https://doi.org/10.1037/0033-2909.125.5.576>

Reviewer #3 (Remarks to the Author):

The article considers three factors that affect demand for interpretability in AI: intentionality, stakes, and scarcity of resources. This is clearly an important setting and the empirical data are clear and thoughtfully interpreted. That said, I had a number of remaining questions after reading the paper:

1. I wonder whether all three factors that the authors study can be subsumed under stakes – more intentional agents (e.g., deciding vs. recommending) and more scarce resources will also presumably be seen as higher stakes settings. Study 2 helps to address this by examining trade-offs in stakes vs. scarcity, but in life these concepts will often positively correlate, making me wonder if the underlying psychology is similar (i.e., people prefer more interpretability for more consequential AI).

2. For the research question examined in Studies 3 & 4, I worry that the empirical results can be summed with a simple thought experiment: Would you rather have AI you understand but doesn't work or AI that works but you don't understand? Presumably, at least for important AI functions, most people would want the latter—and that's what the authors find.

3. Regarding the theory and boundary conditions, I am curious whether the types of stakes matter. The authors defined higher stakes as having larger consequences for human welfare. I wondered if settings that could affect human life (e.g., medical decision, parole) might be treated differently than ones that would be financially costly (e.g., shopping customer service agents). You could argue that both matter for human welfare, but the former are more morally charged and perhaps seen as particularly important/sacred.

4. There seemed to be some missing literature on another factor that might affect demand for interpretability (or accuracy over interpretability) – how much the setting seems to require human expertise or certainty. This effect was shown in Study 3. A few papers that might be relevant:

- Dietvorst, B. J., Bharti, S. (in press). People Reject Algorithms in Uncertain Decision Domains Because They Have Diminishing Sensitivity to Forecasting Error. *Psychological Science*.
- Dietvorst, B. J., Simmons, J. P., & Massey, C. (2015). Algorithm Aversion: People Erroneously Avoid Algorithms After Seeing Them Err, *Journal of Experimental Psychology: General*, 144(1):114-126.
- Logg, J. M., Minson, J.A., & Moore, D.A. (2019). Algorithm Appreciation: People prefer algorithmic to human judgment. *Organizational Behavior and Human Decision Processes*, 151, 90-103.

5. Study 1 Comments

- 5a. I appreciate the large number of stimuli, but would be curious to know how they were selected. In addition, why were the surveillance and virtual assistants missing the "recommend" version?

- 5b. I wouldn't make very much of participants' reporting overall that interpretability was important ($M=3.59$) just because it's quite close to the scale midpoint.

- 5c. How did you determine the stakes and scarcity of each setting? For instance, it is interesting that fraud detection was considered lower stakes than cybersecurity, and that military weapons were not considered scarce. Would be good to re-do this analysis with a set of reliable coders if possible. (I appreciate that the authors mentioned this limitation in their study discussion, but it would be better to rectify.)

- 5d. Just a comment that Figure 1 is amazing! I love all of the information being conveyed here.

- 5e. Study 1 identified another possible predictor of demand for interpretability: applications situated in the public vs. private sphere. I am curious why the authors chose not to pursue this predictor?

6. Study 2 Comments

- 6a. I am curious about why a within-subjects design was selected, given that the scenarios are so parallel. This drives attention toward the differences across conditions and creates a contrast effect. A more conservative test would use a between-subjects design.

- 6b. A significant number of participants (36) were removed for failing an attention check. I wonder if those people would be most likely not to care about interpretability. If so, this suggests that the results could be biased due to participant selection.

- 6c. It seems like participants prefer more interpretability whenever there are more consequences for AI, which I think makes sense. I wonder if participants would simply report preferring anything that

seems like "good" AI –created by more experts, more accuracy, more clarity, easier to use, and so on. They only had the opportunity to answer about interpretability, so we don't really know the extent of the effect here.

7. Study 3 Comments

7a. The instructions presented in this study made me wonder why the authors did not just use the word "understandable" instead of "explainable"?

7b. I found Figure 3 a bit confusing to read – was there a reason why not to split (b) into Figure 3 and (c) into Figure 4? They seem like separate results. Although given that there was no three-way interaction, I don't think (c) is very important to include.

7c. See comment #2 above – these results feel quite intuitive to me.

8. Study 4 Comments

8a. I would really appreciate more detail on why the agency and patency perspectives were tested. Was there any expectation that those perspectives would differ? If not, why were they tested?

8b. Beyond the new factor of agency vs patency, and measuring accuracy and interpretability separately, am I right in concluding that Study 4 is a replication of Study 3? I am not sure if anything new has been learned in Study 4.

9. This was a very well-written Discussion – I enjoyed reading it!

Overall, I thought this paper was quite interesting and I hope to see more work like it from the authors in the future. I hope that some of the feedback above will be useful.

Signed,

Juliana Schroeder

General comments for all Reviewers:

For the sake of readability and brevity, we will refer to studies from our original manuscript as Study 1', Study 2', etc. Meanwhile, Study 1 (A/B/C), Study 2, etc. will denote the studies in the revised manuscript. To summarise, we made the following changes:

Study 1'	=	Study 1A Study 1B (newly added) Study 1C (newly added)
Study 2'	=	Study 2
Study 3'	=	Study 3A (newly added) Study 3B Study 3C (newly added) Study 3D (SI; newly added)
Study 4'	=	moved to SI as Study 4

We have revised all figures to feature information about the distributions of underlying data, which we now provide along measures of central tendency and error bars.

Reviewer #1:

This study considers a very interesting and timely question; human preferences for interpretability in AI-based systems and how that compares to preferences on accuracy. In so doing, the authors present four well designed studies, and provide an appropriate discussion. The study contains various interesting findings that would be very informative and impactful. However, I think there are some crucial issues to address:

1. The main message is not very clear. The paper reads as a collection of interesting results about preferences for interpretability in relation to different factors.

We thank the Reviewer for the helpful prompt to streamline our manuscript. To this end, we have focused the main manuscript on two key findings that characterise attitudes towards interpretability: (i) people value interpretability in AI, especially for applications involving high stakes or scarcity; (ii) even though people care about interpretability in AI across all levels of AI accuracy, they readily sacrifice interpretability for the benefit of accuracy when the two attributes must be traded off.

Our efforts to streamline the paper and address all Reviewers' comments, led us to add four empirical studies that verify the robustness of our findings (Studies 1B, 1C, 3C) and that clarify the relationship between attitudes towards AI interpretability and AI accuracy (Study 3A), while we moved Study 4' to the SI. In summary, the empirical body of our streamlined and revised manuscript proceeds as follows:

First, in Studies 1A-1C, we survey attitudes towards AI interpretability across a wide range of AI applications.

Using an experimental design, Study 2 builds on findings from Studies 1A-1C that point towards *stakes* and *scarcity* as potential driving forces underlying people's valuation of interpretability in AI.

Studies 3A-3C explore how people value AI interpretability across varying levels of AI accuracy and when it trades off with AI accuracy, as is often the case in practice.

2. The most interesting part for me is Study 4, specifically the comparison and interplay between interpretability and accuracy. But even this relationship is not investigated in sufficient depth. This study tells us that people prefer accuracy over interpretability, both when traded off and when taken independently (and regardless of stakes and scarcity). This misses an important and plausible point; preference for interpretability is almost irrelevant for an inaccurate system. An AI that has low accuracy (beyond some threshold) is undesirable regardless of how interpretable it is. It is therefore crucial to investigate the tradeoff in a different way, probably starting from an acceptable-accuracy threshold point rather than from the 'not at all accurate' point. Moreover, demand for interpretability could be dependent on accuracy (but not vice versa). In this case, the question could better be asked as: How strongly would people demand interpretability from an AI that has 100% accuracy? What about an AI with 90% or 80% accuracy? I imagine that demand for interpretability as a function of model accuracy will take a monotonically decreasing shape (after some acceptable-accuracy threshold). But it is an important and an open question of how this would really look like (smooth continuous decrease vs. one big drop), and whether it changes based on the application (medical diagnosing vs. parole reviewing).

We agree that the interplay between interpretability and accuracy is intriguing, not at least in view of our findings from Study 3' (and their replication in Study 4'), but also because of its practical significance.

Hence, with your helpful comment in mind, we ran an additional experiment to clarify how people might value *AI interpretability as a function of AI accuracy* (Study 3A; final $N = 261$). We used the same materials as in Studies 3' and 4', but this time participants indicated their attitudes towards interpretability for different AI models that varied in their accuracy between 60% and 90%. We focused on this accuracy range because there is no reason to use a model that performs merely at chance-level or only slightly better, and few models available to date achieve levels above 90%. The results can be summarised as follows (cited from main manuscript):

"To explore whether participants' attitudes towards AI interpretability were sensitive to variations in AI accuracy, we ran a linear mixed effect model predicting rated importance of interpretability by a fixed effect of accuracy and random intercept effects for subject and application. A type II Wald chi-square test indicated a significant effect of accuracy on interpretability importance, $\chi^2(3) = 11.89, p = .008$, such that participants rated interpretability as less important for AI models with higher accuracy both at the overall level (see Figure 3A) and across all five AI application (see Figure 3B-F). This overall pattern replicated when accounting for various control variables and in particular was not affected by the order in which we presented the AI models varying in accuracy ($p = .422$; see SI Results). Notably, across all levels of accuracy and including the 90% level, participants indicated a high level of

importance for AI interpretability such that their ratings consistently exceeded the 'moderately important' scale-midpoint ($M_s \geq 3.72$; one-sample t -tests yielding $p_s < .001$, Cohen's $d_s \geq .54$).

Our findings from Study 3A indicate that attitudes towards interpretability in AI are remarkably stable across different levels of AI accuracy and that they average at a level valuing AI interpretability consistently as more than “moderately” important. While Study 3A asked participants to evaluate the importance of interpretability across independently varying levels of accuracy, in practice AI interpretability might come at the cost of AI accuracy^{4,5,15,16}. Thus, in our next step we sought to explore how people value AI interpretability when it comes as a tradeoff with AI accuracy.” (pp. 10 f.)

To summarise, the results from our additional experiment confirm your intuition that -- overall -- attitudes for interpretability in AI describe a continuously though only minimally decreasing shape, whereby people are slightly less concerned about AI interpretability for AI models performing with higher accuracy. However, even for highly accurate AI models they still consider interpretability as more than “moderately” important.

3. Participants are recruited from MTurk, which does not provide nationally representative samples. The authors do not discuss this point in the manuscript. How do we know that these results hold for the whole population? Discussing this in the paper is a bare minimum, but I don't think it is sufficient for a publication in this journal. The authors need to show that the results (at least the ones in Study 4) would generalize to the population residing in the US, either by replicating the study through platforms that provide representative samples, or (less preferably) doing some post-stratification or sensitivity analysis.

We agree with you and Reviewer 2 that in characterising *public* attitudes towards interpretability in AI, it is important to consider samples that represent the composition of a population along central variables such as age, gender, and race. To solidify the robustness of our findings in this regard, we ran two pre-registered replications of Study 1' in which we recruited representative samples from the United States and the United Kingdom, respectively, from a different platform (Prolific). The findings, reported in Studies 1B and 1C, replicate a robust yet variable demand for interpretability in AI across the representative samples from both nationalities. Furthermore, using a categorisation of stakes and scarcity that was validated by an independent set of raters, these two variables were corroborated as potential driving forces in people's valuations of interpretability.

Additionally, we replicated our results from Study 3', which you identified as particularly interesting, using a representative US sample and a between-subjects design (see R1_5; Study 3C in the revised manuscript) and we also used a representative US sample for Study 3A. Sample nationalities are now specified explicitly for each study.

Finally, our Discussion now highlights that

“the conclusions drawn from the present work are limited to participants from the US and the UK. Exploring demand for interpretable AI among other populations is a promising and important topic for future work, especially in light of recent work suggesting that expectations towards machine-made decisions can vary substantially across countries and cultures⁴⁷ and amidst reports about a proliferated impact of AI on the lives of low-income populations outside Western, educated, industrialized, rich, democratic contexts⁴⁸.” (p. 16)

4. The findings of this study about preferences is limited to 'stated preferences'. I may have missed it, but I don't think this point is mentioned in the manuscript. So, is the goal of the paper capturing stated preferences? Or does the paper rely on the assumption that stated preferences are not very different from revealed preferences in this context? I understand it's probably not easy (yet not impossible) to devise a study to capture revealed preferences in this case, but the authors need to discuss this point, and either show empirical evidence for this assumption or argue why they think it is sensible in this case. Only acknowledging this as a limitation is not enough.

We think that it is sensible and justifiable to focus on stated preferences in the present work primarily because it is *stated* rather than revealed preferences that are shaping policies governing the use of AI in regards to interpretability and other properties, both at present and also in the foreseeable future. Indeed, most of those in charge of developing policy-

frameworks and most potential users do not have the chance to try out these technologies, let alone different versions of it, before they decide how they feel about it. Thus, measuring stated preferences is the most ecologically valid way to test our research questions. Furthermore, it would have been impossible (and unethical, e.g., for the applications set in legal contexts) to study the range of applications that we examined here using incentivised choices.

That said, we certainly want to be clear about (i) the fact that our work is limited to characterising stated preferences, and (ii) the potential to extend our research towards characterising revealed preferences for interpretability in the future. With this in mind, in the revised manuscript we now refer to “attitudes towards interpretability in AI” rather than “preferences for interpretability in AI” and have extended the discussion as follows:

“As the technical implementation of different degrees of interpretability in AI develops, policy-makers and users alike might update their a priori attitudes towards interpretability in AI. It will then become possible and important for future research to characterise how the findings described in the present work evolve over time, and to explore how attitudes towards interpretability translate into manifest choices⁴⁹. For instance, a scenario conceivable in the near future might be that healthcare providers will offer patients a choice between a version of a medical algorithm that vastly outperforms human doctors in medical diagnosing but that is not at all interpretable³, or a version that performs slightly better than human doctors and that is interpretable. It will be important to characterise people’s revealed preferences in such settings, which will also allow to explore whether, and if so how, valuations of interpretability differ between a priori appraisals, where outcomes are unknown and on which we focused in the present work, as opposed to a posteriori appraisals, where outcomes are known”. (p. 16)

5. This study is limited to American participants (or participants residing in the United States), a point acknowledged in the Discussion (but only there). That the participants are Americans or live in the US should be mentioned at least in the Methods section and in the main manuscript at the beginning of each study when describing participants and N.

We have replicated Study 1’ in a UK sample to improve generalizability, and we now clarify sample nationalities throughout the manuscript and SI.

6. The effect of scarcity and stakes is studied in a within-subject design in Study 2-4. It is never varied between subjects. The authors need to at least show that a carryover effect is unlikely here (which they can do given their order randomization), or (even better) re-run the study in a between-subject design.

We fully agree with your comment, which is also echoed by Reviewers 2 and 3, that a within-subjects manipulation of stakes and scarcity might trigger carryover (or contrast) effects. To verify the robustness of our results, we ran an additional experiment which is reported as Study 3C in the revised manuscript. In this pre-registered experiment, which used a sample representative of the US population in terms of its gender, age, and race composition, we manipulated stakes and scarcity *between* subjects. Put differently, each participant encountered only one combination out of the four possible (low versus high) stakes by (low versus high) scarcity combinations. Using this stronger test, we replicated main effects for

stakes ($p < .001$, $d = .12$) and scarcity ($p = .008$, $d = .03$) on demand for interpretability in AI. However, we also note that effect sizes were smaller relative to the original within-subjects Study 3B (stakes: $d = .12$; scarcity: $d = .09$) and discuss the possibility that effects of stakes and scarcity are influenced by the salience of variation in the two attributes. This interpretation is supported by another new study, that we report as Study 3D in the SI. In this study, we eliminated any information about variability in stakes and scarcity. For instance, in the case of allocating flu vaccines, we would only mention one particular level of vaccine supply (abundant or limited) but not the possible range. In this setting, only the main effect for stakes remained significant ($p < .001$) whereas the effect for scarcity was no longer significant ($p = .141$). We further reflect on these insights from Studies 2-3C as follows in the main manuscript:

“Over time, as the use of AI spreads ever more widely, people will be increasingly likely to encounter variations of stakes and scarcity within and across AI-applications in the real-world. This will arguably enhance people’s sensitivity to stakes and scarcity present in a given AI application and foster the formation of more systematic and stable preferences over accuracy and interpretability in AI³³.” (p. 14)

7. How were the annotations of scarcity and stakes levels in Study 1 done? Was that done by one of the authors or was it done by participants? It seems like it was done by the authors, which would be a problem. I think neither of the two factors is perfectly objective, and they could be open for interpretation. For example, medical diagnosing could be thought of as a low stakes, and automated grading is not likely to be understood as a scarce commodity case (unless you’re grading on a curve). I know that the authors used this as an exploration point which was properly designed for in Study 2, but I still think that the correlational evidence in Study 1 add another type of evidence (more useful for prediction than inference) about tasks that are innately high/low stakes and (not) scarce. If this was not done by the participants, I’d suggest asking 1-3 people who are not familiar with the manuscript findings to do the annotation.

We absolutely agree with your point and (i) further clarified our categorisation of stakes and scarcity in our revised manuscripts, besides (ii) using a validated categorisation in our replications of Study 1’.

In particular, we are now explicit that in Study 1A (corresponding to Study 1’) we used an exploratory post-hoc categorisation that was performed by two of the authors: *“To this end, two of the authors performed a hand-coded categorisation of stakes and scarcity involved in a given application after data collection was complete”* (pp. 5 f.). Additionally, we ran two pre-registered replications of Study 1’ for which we used a validated categorisation of applications (Studies 1B and 1C). To validate our categorisation, we asked nine independent raters (i.e., research assistants who were not involved nor familiar with the present work or its findings) to categorize each application according to the involved stakes (low/medium/high) and scarcity (yes/no). Aggregating across vignettes, raters agreed in their stakes categorizations 70% of the time, and 84% of the time in their scarcity categorizations. Using the median responses of these independent raters’ categorisations, we replicated main effects of stakes ($p < .001$) and scarcity ($p < .001$), as reported in Studies 1B and 1C of the revised manuscript (pp. 7 f.).

Reviewer #2:

This paper presents four studies investigating public demand for interpretable AI. In these studies, participants from an online panel rated how important they thought it was for AIs to be interpretable in various applications, or their preferred tradeoff between AI interpretability and accuracy. The results show that people generally think it is important for AIs to be interpretable, especially when the stakes of the decision are high and when the decision involves a scarce resource. However, when contrasted to accuracy, people showed a greater preference for accuracy over interpretability for decisions involving high-stakes and scarce resources. The perspective participants were asked to take – that of a person affected by the decision or of a person responsible for managing the AIs decision – did not affect people’s preference for accuracy over interpretability.

To the best of my knowledge, the paper is novel. Indeed, as the authors mention, questions of AI interpretability have been discussed, but there are no systematic investigations of public demand of interpretable AIs.

A systematic investigation of public demand for interpretable AIs could indeed be of interest to many in the field of human-robot interaction. Although the paper in its current form does not significantly advance theory in the field, a systematic test of whether laypeople show the same considerations engineers and ethicists discuss is important.

Several aspects of the paper can be strengthened. I expand on several suggestions below. Some of these suggestions are more difficult to implement than others. These should be viewed as suggestions, and not all of them should be seen as necessary.

1. The paper could benefit from more diversity in the measurement of “demand for interpretable AI”. Currently, two ways are used to measure “demand”. 1) self-reported “importance” (studies 1-2); 2) self-reported preference for the interpretability-accuracy tradeoff (studies 3-4). I am not sure that either of these measures necessarily reflects “demand”, at least not in the economic sense. “Importance” or “preference” might be more accurate terms. The authors can strengthen the paper by expanding its measurement. For example, by examining how people rely on various AIs on the interpretable-accurate tradeoff for a joint task. Another possibility is to explore the market response to various AI systems that vary in their interpretability and accuracy. Are certain systems more popular than others? These are just a few examples, I am sure there are many more. I do think, however, that the current measurement needs strengthening.

We appreciate that using the term “*demand* for interpretable AI” might not be an ideal reflection of the measures presented in the original submission and that these were limited to two varieties (importance- and tradeoff-ratings). The other two reviewers also echoed this point in slight variations. In response, we sought to clarify our focus on “attitudes towards interpretability in AI”, rather than demand for interpretability in terms of revealed choices. We explain and justify our focus as follows:

“We focus on characterising public attitudes towards AI interpretability rather than revealed choices, because the present debates about interpretable AI take place prior to widespread technological development or deployment of AI systems systematically

varying in terms of interpretability. Hence, public attitudes – rather than revealed preferences – seem critical for policy development at present.” (p. 3)

Our revised discussion additionally emphasises our contribution's focus on attitudes towards AI interpretability and highlights that it may be an interesting and worthwhile avenue to follow how those attitudes evolve over time and how they translate into manifest choices as AI systems varying in interpretability become more available and prevalent:

“As the technical implementation of different degrees of interpretability in AI develops, policy-makers and users alike might update their a priori attitudes towards interpretability in AI. It will then become possible and important for future research to characterise how the findings described in the present work evolve over time, to explore how attitudes towards interpretability translate into manifest choices, and how they relate to attitudes towards explanations of human decisions⁵⁰. For instance, a scenario conceivable in the near future might be that healthcare providers will offer patients a choice between a version of a medical algorithm that vastly outperforms human doctors in medical diagnosing but that is not at all interpretable³, or a version that performs slightly better than human doctors and that is interpretable. It will be important to characterise people's revealed preferences in such settings, which will also allow to explore whether, and if so how, valuations of interpretability differ between a priori appraisals, where outcomes are unknown and on which we focused in the present work, as opposed to a posteriori appraisals, where outcomes are known.” (p. 16).

But we also sought to address your concern by running additional studies. In particular, Studies 1B and 1C in the revised manuscript sought to ensure that our findings from Study 1' were robust to varying the used terminology. Indeed, we replicated overall positive but also variable attitudes towards interpretability in AI using a probe that referred to “*understandable AI*”.

2. As the paper sets out the measure “public demand”, a more comprehensive sample (or samples) could also strengthen the argument about “public”. Relying on one online platform and using relatively small samples limits generalizability. This is especially important since the goal of the paper is to investigate “public demand”. Although this limitation is mentioned in the discussion, using larger samples, collected on multiple platforms, could strengthen the paper and the argument it makes. This will also allow exploring individual and cultural differences.

We fully agree with your comment that characterising “public” attitudes towards interpretability in AI requires samples that adequately reflect the composition of a public audience. We sought to validate our results in this regard by conducting three follow-up studies with larger, representative sampling procedures that were implemented on different platforms (*viz.*, Studies 1B, 3B, 1C, 3A). In particular, for those follow-up studies we used Prolific Academic as a different recruitment platform (previous samples had been recruited on MTurk) and collected samples that were representative of the US population in terms of its gender, age, and race. For Study 1C, we recruited a representative UK sample which allowed us to verify that our findings generalise from an American population (Studies 1A and 1B) to another Western sample.

We also wholeheartedly agree that exploring cultural variation in attitudes towards interpretability is an important task. However, we do think that it would go beyond the scope of the present work to attempt this. Hence, we explicitly emphasise this limitation of our work and call for future research to address this gap.

“Furthermore, the conclusions drawn from the present work are limited to participants from the US and the UK. Exploring attitudes towards interpretable AI among other populations is a promising and important topic for future work, especially in light of recent work suggesting that expectations towards machine-made decisions can vary substantially across countries and cultures⁴⁹ and amidst reports about a proliferated impact of AI on the lives of low-income populations outside Western, educated, industrialized, rich, democratic contexts⁵⁰.” (p. 16)

3. One reason why the idea of AI interpretability became popular, is that it is seen as a way to safeguard from discrimination (Gilpin et al., 2019). Interpretability is sometimes thought of as a way to create responsible, accountable and fair AI systems. Of course, one paper cannot examine all possible factors. However, another way to further strengthen the paper is to explore an interpretability-fairness tradeoff.

We agree that safeguarding against discrimination is a plausible and widely debated motivation for requiring interpretability in AI. Already in our original manuscript, our introduction highlighted the fact that interpretability could be a key factor in creating fair systems:

*“A third aspect of AI applications that might drive attitudes towards interpretability is its potential gatekeeping function. Many emerging AI applications are designed to determine access to scarce but desirable resources, such as jobs, financial loans, or medical care. Ample theoretical and empirical work demonstrates that people demand explanations for decisions involving the allocation of resources²⁷⁻²⁹, **especially when those resources are scarce^{30,31}, in order to ensure that the allocation procedure was fair.**” (p. 4; emphasis added)*

Based on these considerations, we moved on to empirically study whether and to what degree people’s fairness concerns drive variation in people’s attitudes towards AI interpretability. Studies 1A-1C provided correlational evidence suggesting that fairness concerns can indeed explain some of the variation that we observed in people’s valuation of AI interpretability; in Studies 2, 3B, and 3C an experimental manipulation of the scarcity involved in a given AI application confirmed that under conditions where fairness concerns are more salient (i.e., under high scarcity), interpretability is valued more strongly.

Our manuscript further discusses the role of interpretability as a safeguard for fairness in the discussion, including the paper by Gilpin and colleagues from 2019, and additional ones such as Kleinberg et al., 2018, which have proposed that interpretability is essential for detecting discrimination:

“[...], reflecting interpretability’s essential capacity of verifying appropriate and fair decision processes^{7,27,42,43}.” (p. 15)

“These findings build on past work showing that people demand more explanation for decisions involving resource allocation in order to ensure that the allocation process was fair^{11,43,44}, demonstrating that such principles also operate in the context of AI applications and substantiating calls for interpretability as a safeguard for ethical and fair AI systems.” (p. 15)

Thus, we are confident that our work highlights and confirms the importance of interpretability in AI as a safeguard against discrimination, and in support of fair decisions. Of course, more work remains to be done to explore how interpretability could best cater to this purpose. While we think that exploring this avenue would go beyond the scope of our work, we hope that it contributes to inspiring such follow-up research.

4. Another way to further strengthen the paper is by uncovering the psychological mechanisms underlying the demand for interpretability. Why do people want interpretability? Why do stakes and scarcity affect the importance of interpretability? Is there a psychological difference between the importance of understanding a decision by an AI and a decision by a human agent? Are there cases where people insist on interpretability even at the cost of reduced accuracy? Addressing any of these questions can enhance our understanding of the phenomenon and the contribution of the paper to theory.

These questions are indeed intriguing and call for a substantive research programme on the psychology underpinning people’s attitudes towards interpretability in AI. Our work seeks to provide a first step in this direction: across seven empirical studies reported in the main manuscript, and two additional studies reported in the SI, we characterise public attitudes towards AI interpretability across a wide range of applications, and – directly addressing the first two questions you raise here – demonstrate a causal role for stakes and scarcity in how much people value interpretability in AI. Both our introduction and our discussion sections tie stakes and scarcity to existing theoretical accounts as well as empirical evidence from decision contexts outside of AI settings, where it has been shown that people value explanation more when outcomes are larger in magnitude or more scarce in terms of availability. Furthermore, both our introduction and overall discussion elaborate on the point that people might care more about explanation in such settings, because it facilitates understanding and guides subsequent learning, prediction, and feelings of control – all of which are particularly important when stakes are high, or when resources are scarce. Take, for example, the following passage has remained the same as in the original manuscript:

“Because interpretability plays a crucial role for predicting, auditing, and controlling underlying decision-making processes^{24,27}, it should be particularly important in settings where AI has large consequences for human welfare. Considering low- versus high-stake cases within the same domain reinforces this intuition: you would probably care more about understanding why an AI accepted or rejected your application for a salaried permanent job, compared to an unpaid honorary job. Indeed, decades of research have documented that people demand explanations more for high-stakes than low-stakes decisions^{7,13,24}.” (p. 4)

Our original manuscript also spoke to the question whether people, in some cases, insist on interpretability at the cost of reduced accuracy: Study 3’ suggested that it is rather the other

way round with participants showing a tendency to prioritize accuracy at the expense of interpretability across AI applications. The revised manuscript provides a replication of these findings using a between-subjects design (Study 3C), as well as an extension showing participants seem to value interpretability across all levels of AI accuracy, that is even for AI performing highly accurately, when the two attributes do not come as a tradeoff (Study 3A). Comparing AI-made versus human-made decisions certainly is another interesting avenue, as further highlighted by recent work on medical AI (Cadario et al., 2021; ref. 50 in the revised manuscript), that deserves an entire series of experiments in its own right.

As you noted yourself, “there are no systematic investigations of public demand of interpretable AI” to this date. It is from this starting point that we embarked with our contribution, which – acknowledging the limits of our contribution being a single paper rather than an entire research programme – we think covers substantial theoretical territory and provides empirical insights that will hopefully inspire further research.

5. All of the experiments used a within-subject design. Such designs can indeed increase the salience of some attributions (Hsee et al., 1999). The paper could be strengthened by examining the pattern of preferences also in a between-subject design. Manipulating AIs within-subjects can be easily justified, as people might need to choose between two AIs, one more accurate and one more interpretable. Regardless, the paper should justify the choice of experimental design and its advantages and limitations.

We fully agree with your comment, which was also echoed by Reviewers 1 and 3. To verify the robustness of the key experimental results that used a within-subjects design, we ran an additional experiment which is reported as Study 3C in the revised manuscript and that followed your suggestion in manipulating stakes and scarcity between subjects. In this pre-registered experiment, which used a sample representative of the US population in terms of age, race, and gender, each participant encountered only one combination out of the four possible (low versus high) stakes by (low versus high) scarcity combinations. Using this stronger test, we replicated main effects for stakes ($p < .001$, $d = .12$) and scarcity ($p = .008$, $d = .03$) on attitudes towards AI interpretability. However, we also note that effect sizes are smaller relative to Study 3B (stakes: $d = .12$; scarcity: $d = .09$) and, drawing on Chris Hsee’s evaluability hypothesis, discuss the possibility that effects of stakes and scarcity are influenced by the salience of variation in the two attributes. This interpretation is supported by another new study, that we report as Study 3D in the SI. In this study, we eliminated any information about variability in stakes and scarcity. For instance, in the case of allocating flu vaccines, we would only mention one particular level of vaccine supply (abundant or limited) but not the possible range. In this setting, only the main effect for stakes remained significant ($p < .001$) whereas the effect for scarcity was no longer significant ($p = .141$). We further reflect on these insights from Studies 2-3D as follows:

“Over time, as the use of AI spreads ever more widely, people will be increasingly likely to encounter variations of stakes and scarcity within and across AI-applications in the real-world. This will arguably enhance people’s sensitivity to stakes and scarcity present in a given AI application and foster the formation of more systematic and stable preferences over accuracy and interpretability in AI³⁴.” (p. 14)

6. Interpreting the null effect for perspective in Study 4 should state more clearly the limitations of not finding a difference with NHST (Null Hypothesis Significance Testing). The authors can strengthen the paper by reporting the statistical power of their experiment or, preferably, by using a Bayesian approach.

We agree that this inherent limitation of NHST constrains the interpretation of Study 4' and we make this limitation more explicit in our revised submission. Given that the exploration of different stakeholder perspectives has not been a central focus of our work and, as noted by Reviewer 3, makes only an incremental contribution to Study 3', we have moved Study 4' into the SI, where it is reported as Study 5.

7. Including the full study materials in the supplemental materials would be very helpful. This should include the full studies and the exact text of all collected variables. Reading the paper, it was sometimes unclear exactly how the measurement was done, and the methods section or the SI did not provide enough detail. For the sake of transparency, I would suggest clarifying whether other variables were collected as well. As the studies were not pre-registered, such a clarification would be helpful, and make the studies easier to reproduce.

The originally submitted SI had already included all vignettes and also screenshots of the dependent variables for Studies 1' and 2' (those for Studies 3' and 4' were shown in the main manuscript). That said, we took care to include all key materials both in the revised SI and in the pre-registrations of our follow-up experiments (Studies 1B, 1C, 3A, 3C). For further transparency, we will also share PDF printouts of our full surveys.

8. Similarly, describing the statistical analysis in more detail could be helpful. Some analyses (such as Study 3, in lines 243-248) have more detail than others. Adding the full details of all reported analyses could be helpful. In some cases, adding justifications for the statistical approach could be beneficial as well. For example, explaining why sometimes an ordinal regression was used and sometimes a linear regression.

We apologise that some details were missing in the original submission. Our revised manuscript now details the rationale for using ordinal regression (*"Because participants gave their answers on a discrete rating scale, we used mixed effect ordinal regression analysis with a fixed effect for condition and a random intercept effect for subject."*, p. 5). We also explain that we used separate models for estimating effects of stakes and scarcity in Studies 1A-1C because of multicollinearity: stakes and scarcity covaried in the sense that almost all applications involving high scarcity also involved high stakes. This limitation is then addressed in Study 2.

To further improve the consistency and robustness of our statistical analysis, we now use mixed effect (rather than simple) multinomial regression analysis for Study 1A, accounting more appropriately for the data being nested within participants (ICC = .20). Results did not change substantially even though the previously observed small difference whereby interpretability is rated as less important for AI systems making recommendation rather than decisions ($p = .015$ for simple regression) is no longer significant ($p = .189$ for mixed effect regression). Fixed and random effects are explicitly specified for each results section.

9. Finally, pre-registration and discussion of statistical power are becoming standard in experimental psychology. I understand that it might be a little late for pre-registration. However, I recommend pre-registering any additional study that might be conducted.

We very much agree that pre-registration is a worthwhile effort towards making science more open and transparent. We have pre-registered all four follow-up studies (Studies 1B, 1C, 3A, 3C; Study 3D reported in the SI) and provide measures of effect size along with confidence intervals throughout our revised manuscript.

Reviewer #3:

The article considers three factors that affect demand for interpretability in AI: intentionality, stakes, and scarcity of resources. This is clearly an important setting and the empirical data are clear and thoughtfully interpreted. That said, I had a number of remaining questions after reading the paper:

1. I wonder whether all three factors that the authors study can be subsumed under stakes – more intentional agents (e.g., deciding vs. recommending) and more scarce resources will also presumably be seen as higher stakes settings. Study 2 helps to address this by examining trade-offs in stakes vs. scarcity, but in life these concepts will often positively correlate, making me wonder if the underlying psychology is similar (i.e., people prefer more interpretability for more consequential AI).

We thank you for this thought-provoking argument. We agree that it is conceivable that situations involving less human agency might be perceived as more consequential, as might situations involving scarce resources. However, it is also conceivable and indeed an empirical reality that cases where these concepts are not positively correlated exist. Our findings highlight that people do distinguish between these two situational features when appraising the value of interpretability and accuracy in AI – not only in settings where they are confronted with direct contrasts in the two attributes (as in the within-subjects design presented in our original submission) but also in a setting presented in Study 3C of the revised manuscript, where we manipulated stakes and scarcity as between-subjects factors. Together, these findings robustly demonstrate dissociable main effects for stakes and scarcity, indicating that people do distinguish between these two situational features when appraising the value of interpretability and accuracy in AI. Moreover, they seem to show a consistent weighting of the two aspects, with stakes consistently having a stronger influence on attitudes towards interpretability than scarcity. We think that this additional nuance is helpful in deciding when interpretability in AI is particularly important, and when it might be less important. Indeed, many features might make a decision more “consequential” – not only the stakes and scarcity involved, but also the demographics of an affected population, the number of affected people, and so on. While the underlying psychology might be similar and amount to a global appraisal of how consequential a decision is, we believe it is still informative and hence worthwhile to attempt delineating individual driving forces and determining their relative influence.

In view of many calls for AI interpretability emphasising its potentially crucial role for predicting, auditing, and controlling underlying decision-making processes, which ultimately aims to ensure fair and just decisions, and in view of existing psychological literature that has highlighted stakes and scarcity as important factors in appraisals of procedural justice, our present work focused on these two factors. That said, we consider our work not as a comprehensive assessment of non-experts’ valuation of interpretability in AI, but only as a starting point for an empirical approximation of non-experts’ valuation of interpretability in AI that deserves more attention. We sought to highlight this limitation in our revised manuscript:

“Given the lack of empirical work on people’s valuation of interpretability in AI, our contribution started by considering two variables – stakes and scarcity – that have emerged as important from existing literature on appraising the “good-“ and “fairness” of human-made decisions. But the nature and scale of machine-made decisions warrants for considering further features of decision situations and their role in

people's valuation of interpretability^{50,52}. For instance, in view of recent work indicating that perceptions about required human expertise in a given decision-context affect people's willingness to rely on algorithmic advice in that context⁵¹, we explored 'required human expertise' as an additional explanatory variable and find tentative evidence (see SI Results) that perceiving high expertise-requirements might sway tradeoff-preferences away from prioritising accuracy, towards affirming the importance of interpretability. Similarly, an important avenue for future research will be to further characterise the relationship between different features of AI systems. Our results from Studies 3A-C indicate that while people have dissociable preferences for individual AI features, complex patterns and interactions emerge when technological constraints, such as interpretability-accuracy tradeoffs, are considered. While the present work reflects the focus on AI interpretability and accuracy as hallmarks of desirable AI, we hope that it generates further research into other AI properties such as transparency or usability." (pp. 16 f.)

2. For the research question examined in Studies 3 & 4, I worry that the empirical results can be summed with a simple thought experiment: Would you rather have AI you understand but doesn't work or AI that works but you don't understand? Presumably, at least for important AI functions, most people would want the latter—and that's what the authors find.

Thank you for this comment which led us to revisit our previous results on the relationship between people's valuation of AI interpretability and AI accuracy and results from an additional experiment (Study 3A) from the revised submission. The results from the original Study 3' (3B in the revised manuscript) and its replication (Study 3C) generally support your conjecture that people trend towards preferring "AI that works but you don't understand" over "AI you understand but doesn't work". We believe that the intuitiveness or simplicity of this insight does, however, not reduce its significance. Furthermore, our work provides a more fine-grained perspective on people's attitudes towards interpretability-accuracy tradeoffs: facing a direct tradeoff between the two attributes, our participants leaned towards prioritizing accuracy over interpretability. That said, their importance ratings scored closer to the point of balance than to the accuracy-extreme. Additional nuance comes from our observation that there are factors (i.e., stakes and scarcity) that push those tradeoff-preferences around. We revised what was previously Figure 3 to illustrate these points more accessibly (Figure 4 of the revised manuscript, which is pasted in our response to your comment #7.2).

Finally, an additional experiment reported in Study 3A highlights that when participants consider AI interpretability in relation but not in inverse dependency to AI accuracy, they value AI interpretability as more than "moderately" important across all levels of AI accuracy between 60% and 90%.

In summary, we believe that the findings summarised in our manuscript offer relevant starting points to empirically solidify conjectures about public attitudes towards interpretability, which are becoming increasingly relevant in developing AI technologies and the surrounding policy regulations.

3. Regarding the theory and boundary conditions, I am curious whether the types of stakes matter. The authors defined higher stakes as having larger consequences for human

welfare. I wondered if settings that could affect human life (e.g., medical decision, parole) might be treated differently than ones that would be financially costly (e.g., shopping customer service agents). You could argue that both matter for human welfare, but the former are more morally charged and perhaps seen as particularly important/sacred.

Thank you for the opportunity to clarify our thoughts regarding these boundary conditions. In some sense, all applications that we studied matter for human welfare -- including the ones that involve financial aspects. As such, 'human welfare' seems too broad as a category to study. Instead, we focused on low versus high stakes involved. Our rationale for this focus was informed by a broad literature demonstrating that human cognition is fundamentally sensitive to variations in the magnitude of outcomes; from economic to moral ones (*"Notably, this sensitivity to stakes parallels magnitude-sensitivity as a foundational process in the cognitive appraisal of outcomes"*^{40,41}, p. 15). Of course, outcomes or stakes may be partitioned along dimensions such as "financial" or "moral" independently from magnitude. However, one could define arbitrarily many possible such types of stakes – human welfare and financial ones, moral and nonmoral ones; but also tangible and intangible ones, nonsocial and social ones, etc. Our choice to test for variations in attitudes towards interpretability along the coarser distinction of low versus high stakes avoids this challenge and allows for more generalisable insights. Additionally, it relates our work more directly to existing psychological research on the function of explanation in human-made decisions (e.g., Lombrozo, 2006): also for AI-made decisions, we find that people value explanation as more important in high-stakes settings. Importantly, we also observe that fairness concerns -- representing an integral aspect of morality -- drive variation in attitudes towards AI interpretability robustly and fully independently from stakes. Again, this ties in with ample evidence spanning literatures from moral decision-making to distributive justice and which has demonstrated that people show enhanced demand for explanation in decisions involving the allocation of scarce resources. We extended our discussion to further reflect upon this point and its ethical implications:

"These findings build on past work showing that people demand more explanation for decisions involving resource allocation in order to ensure that the allocation process was fair^{11,43,44}, demonstrating that such principles also operate in the context of AI applications and substantiating calls for interpretability as a safeguard for ethical and fair AI systems. Enhanced valuation of interpretability in such settings seems all the more justified and important in view of recent anecdotal evidence that (apparent) lack of interpretability may provide human agents in charge of overseeing outcomes produced by AI systems with the opportunity to obscure personal responsibility: when allocation decisions for vaccines against Covid-19 went awry, prioritising administrators before frontline healthcare workers, responsible officials readily blamed a "very complex algorithm" for the undesirable outcomes⁴⁵. The fact that this algorithm turned out to be a relatively simple and hand-coded rule-based formula⁴⁶ highlights the danger that humans in charge may purport lack of interpretability in AI even when this is not the case." (p. 15)

We hope that our clarifications and the addition to our discussion address your concern, as much as we hope that future research will provide more fine-grained analyses for the role of other aspects of morality in people's attitudes towards AI interpretability.

4. There seemed to be some missing literature on another factor that might affect demand for interpretability (or accuracy over interpretability) – how much the setting seems to require human expertise or certainty. This effect was shown in Study 3. A few papers that might be relevant:
 - Dietvorst, B. J., Bharti, S. (in press). People Reject Algorithms in Uncertain Decision Domains Because They Have Diminishing Sensitivity to Forecasting Error. *Psychological Science*.
 - Dietvorst, B. J., Simmons, J. P., & Massey, C. (2015). Algorithm Aversion: People Erroneously Avoid Algorithms After Seeing Them Err, *Journal of Experimental Psychology: General*, 144(1):114-126.
 - Logg, J. M., Minson, J.A., & Moore, D.A. (2019). Algorithm Appreciation: People prefer algorithmic to human judgment. *Organizational Behavior and Human Decision Processes*, 151, 90-103.

We thank you for flagging these references, which we included in our revised submission along with a recent paper by Cadario and colleagues (2021), which suggests that differences in subjective understanding of human- versus machine-made decisions might translate into diverging requirements for explanation towards the respective agent.

Additionally, the revised discussion explicitly points out the effect of expertise, which will hopefully attract more attention in future research:

“Given the lack of empirical work on people’s valuation of interpretability in AI, our contribution started by considering two variables – stakes and scarcity – that have emerged as important from existing literature on appraising the “good-“ and “fairness” of human-made decisions. But the nature and scale of machine-made decisions warrants for considering further features of decision situations and their role in people’s valuation of interpretability^{50,52}. For instance, in view of recent work indicating that perceptions about required human expertise in a given decision-context affect people’s willingness to rely on algorithmic advice in that context⁵¹, we explored ‘required human expertise’ as an additional explanatory variable and find tentative evidence (see SI Results) that perceiving high expertise-requirements might sway tradeoff-preferences away from prioritising accuracy, towards affirming the importance of interpretability. Similarly, an important avenue for future research will be to further characterise the relationship between different features of AI systems. Our results from Studies 3A-C indicate that while people have dissociable preferences for individual AI features, complex patterns and interactions emerge when technological constraints, such as interpretability-accuracy tradeoffs, are considered. While the present work reflects the focus on AI interpretability and accuracy as hallmarks of desirable AI, we hope that it generates further research into other AI properties such as transparency or usability.” (pp. 16 f.)

5. Study 1

- 5.1 I appreciate the large number of stimuli, but would be curious to know how they were selected. In addition, why were the surveillance and virtual assistants missing the “recommend” version?

In our revised manuscript, we highlight that “*We compiled our collection of AI applications by surveying newspaper articles, technological reports, and scientific papers, with the aim of covering a diverse range of applications already in use as comprehensively as possible (see Figure 1 for an overview; see SI Materials for full list of applications with source links and study descriptions)*” (p. 5).

Two key objectives in creating the parallel ‘decide’ and ‘recommend’ versions for Study 1’ was that the AI would be executing exactly the same task for both versions and that these tasks would make sense. We could not not think of a vignette that matched these objectives for the ‘surveillance’ and ‘virtual assistants’ applications:

Decide: AI interacts with a person through speech-recognition-based virtual assistants such as Amazon’s Alexa, Apple’s Siri, or Google Home -- e.g., by turning on the music when the person tells them to do so.

Recommend: Recommending an interaction with itself just does not make sense (alternatively, the AI could recommend things such as turning up the volume but that would change the AI’s task).

Decide: AI surveils suspected criminal offenders through CCTV on behalf of law enforcement authorities.

Recommend: AI recommends to law enforcement authorities which suspected criminal offenders to surveil through CCTV. [changes task from observation to identification]

We also clarify that our analyses for the effect of condition included only response data from those applications, which existed in both versions (p. 5). Results do not change significantly if we include the cases where data from the recommend version was missing. Hence, our conclusions for Study 1’ were not affected by these asymmetries.

5.2 I wouldn’t make very much of participants’ reporting overall that interpretability was important (M=3.59) just because it’s quite close to the scale midpoint.

We agree that this observation is not a key finding and we sought to not present it as such in the revised manuscript. That said, we chose to keep reporting participants’ overall attitudes towards interpretability because they did differ *significantly* from the scale midpoint across all studies that examined attitudes towards interpretability. We think this is noteworthy given that this midpoint was labeled as “moderately important” and hence already exceeding a point of neutrality (e.g., “neither important nor unimportant”). Of course, statistical significance by itself does not justify devoting space to a given finding, but considering the scale-midpoints’ meaning and considering that this overall average also includes low-stakes/scarcity applications, such as picture processing tools or virtual assistants, we think that our decision to report participants’ overall attitudes is warranted. On that note, we would like to mention that we adjusted the reported stats for Study 1’ to reflect the dichotomous nature of the response scale.

5.3 How did you determine the stakes and scarcity of each setting? For instance, it is interesting that fraud detection was considered lower stakes than cybersecurity, and that military weapons were not considered scarce. Would be good to re-do this analysis with

a set of reliable coders if possible. (I appreciate that the authors mentioned this limitation in their study discussion, but it would be better to rectify.)

We absolutely agree with this point and do now clarify that for Study 1A we used a post-hoc categorisation that was performed by two of the authors (*“To this end, two of the authors performed hand-coded categorisations of stakes (“low”, “medium”, “high”) and scarcity (“no”, “yes”;* see Figure 1) involved in a given application after data collection was complete.”; p. 5). Additionally, we ran two pre-registered replications of Study 1’ for which we used a validated categorisation of applications. To this end, we asked nine independent raters (i.e., research assistants who were not involved nor familiar with the present work or its findings) to categorize each application according to the involved stakes (low/medium/high) and scarcity (yes/no). Aggregating across vignettes, raters agreed in their stakes categorizations 70% of the time, and 84% of the time in their scarcity categorizations. Using the median responses of these independent raters’ categorisations, we replicated main effects of stakes ($ps < .001$) and scarcity ($ps < .001$) as reported in Studies 1B and 1C of the revised manuscript.

5.4 Just a comment that Figure 1 is amazing! I love all of the information being conveyed here.

We very much appreciated this positive feedback!

5.5 Study 1 identified another possible predictor of demand for interpretability: applications situated in the public vs. private sphere. I am curious why the authors chose not to pursue this predictor?

Our main concern was that many applications could easily be transferred from one sphere to the other, while we also wanted to avoid applications for which the line between public and private is blurry. These concerns could be potentially be addressed by creating a set of vignettes more detailed regarding the sphere (e.g., specifying whether healthcare provision is public or private), which in turn would require yet another study. Amidst space constraints and in the interest of streamlining our collecting of findings, we decided to not further pursue this avenue in the present work that will hopefully inspire future work picking up on this finding.

6. Study 2

6.1 I am curious about why a within-subjects design was selected, given that the scenarios are so parallel. This drives attention toward the differences across conditions and creates a contrast effect. A more conservative test would use a between-subjects design.

We fully agree with your comment, which is also echoed by Reviewers 1 and 2, that a within-subjects manipulation of stakes and scarcity might trigger carryover (or contrast) effects. To verify the robustness of our results in this regard, we ran an additional experiment which is reported as Study 3C in the revised manuscript. In this pre-registered experiment, which used a sample representative of the US population in terms of age, race, and gender, we manipulated stakes and scarcity between subjects. Put differently, each participant encountered only one combination out of the four possible (low versus high) stakes by (low versus high) scarcity combinations. Using this stronger test, we replicated main effects for stakes ($p < .001$, $d = .12$) and scarcity ($p = .008$, $d = .03$) on demand for interpretability in AI. However, we also note that effect sizes are smaller relative to Study 3’ (stakes: $d = .12$; scarcity: $d = .09$) and discuss the possibility that effects of stakes and scarcity are influenced

by the salience of variation in the two attributes. This interpretation is supported by yet another study, that we report as Study 3D in the SI. In this study, we eliminated any information about variability in stakes and scarcity. For instance, in the case of allocating flu vaccines, we would only mention one particular level of vaccine supply (abundant or limited) but not the possible range. In this setting, only the main effect for stakes remained significant ($p < .001$) whereas the effect for scarcity was no longer significant ($p = .141$). We further reflect this limitation of our findings from Studies 2-3C as follows:

“Over time, as the use of AI spreads ever more widely, people will be increasingly likely to encounter variations of stakes and scarcity within and across AI-applications in the real-world. This will arguably enhance people’s sensitivity to stakes and scarcity present in a given AI application and foster the formation of more systematic and stable preferences over accuracy and interpretability in AI³³.” (p. 14)

6.2 A significant number of participants (36) were removed for failing an attention check. I wonder if those people would be most likely not to care about interpretability. If so, this suggests that the results could be biased due to participant selection.

Thank you for the opportunity to clarify this point. As specified in the manuscript, these participants did not fail attention checks, but *comprehension* checks. In particular, they were not able to recognise our definition of ‘explainable AI’, or to indicate their task was to rate the importance of explainability of an AI that performs accurately. Because the interpretability of participant-responses depended crucially on (i) them understanding the key concept (explainability) and on (ii) them understanding their task was to rate explainability’s importance, while assuming the AI would perform accurately, we had decided to exclude participants failing these comprehension checks. We acknowledge that it is unfortunate that we did not pre-register the study and hence this a priori exclusion criterion (something that we have rectified in our pre-registered follow-up studies for the revised resubmission).

While we think that this selection procedure is justified, we checked our results’ robustness to including the participants who failed comprehension checks (total $N=120$). We replicated main effects for stakes ($b = 0.69$, $p < .001$, $d = 0.32$, 95% CI [0.24, 0.39]) and scarcity ($b = 0.40$, $p < .001$, $d = 0.18$, 95% CI [0.11, 0.25]) using the random intercept model from the analysis reported in the main manuscript (fixed effects for ‘stakes’ (low versus high), ‘scarcity’ (low versus high), and their interaction along with random intercept effects for ‘subject’ and ‘application’). Again, these main effects were not qualified by an interaction ($p = .886$).

6.3 It seems like participants prefer more interpretability whenever there are more consequences for AI, which I think makes sense. I wonder if participants would simply report preferring anything that seems like “good” AI –created by more experts, more accuracy, more clarity, easier to use, and so on. They only had the opportunity to answer about interpretability, so we don’t really know the extent of the effect here.

Thank you for this interesting point. Our results from Study 4’ (now reported as Study 5 in the SI) and Study 3A indicate that preferences for interpretability and accuracy, which can be considered as two key hallmarks of “good” AI, are dissociable. In line with your conjecture, people seem to prefer more goodness on each dimension when it is considered separately. However, we hope that our paper highlights that the technological constraints of AI systems,

which often require tradeoffs between the two properties, introduce more complex attitude-patterns. Of course, this very observation might point towards the need of exploring and relating more dimensions of “AI goodness” against one another. More generally, we think that your point highlights the value of our paper as a starting point of what could become an entire research programme on the psychology of explainable AI. That said, it is of course not feasible to attempt addressing them all in a single paper. But your and the other reviewers’ comments leave us hopeful that our paper is generative even with its focus on interpretability, which reflects that much of the scholarly and societal attention surrounding “good” AI is, at this point, focused on explainability and how it might be reconciled with technological constraints. We extended our manuscript’s discussion to explicitly highlight these considerations:

“Similarly, an important avenue for future research will be to further characterise the relationship between different features of AI systems. Our results from Studies 3 A-C indicate that while people have dissociable preferences for individual AI features, complex patterns and interactions emerge when technological constraints, such as interpretability-accuracy tradeoffs, are considered. While the present work reflects the focus on AI interpretability and accuracy as hallmarks of desirable AI, we hope that it generates further research into other AI properties such as transparency or usability.”
(p. 17)

7. Study 3

7.1 The instructions presented in this study made me wonder why the authors did not just use the word “understandable” instead of “explainable”?

The terms “interpretable” and “explainable” are most commonly (and often interchangeably) used in the emerging literature on AI interpretability or explainability (e.g., Adadi & Berrada, 2018; Gunning et al., 2019; Murdoch et al., 2019). As argued in our manuscript, we used the term “explainable” for participant instructions because it is more intuitive than “interpretable” and still aligns with the common scientific terminology. While the latter is not true for the term “understandable”, it may be even more intuitive, as you propose. In any case we expect our results to be robust to variations in this terminology, as long as the same definition can be applied. To test this conjecture, our newly added and pre-registered Studies 1B and 1C used the term “understandable” instead of “explainable”. Correspondingly, the introductory instructions and the probe of participants’ attitudes towards interpretability in AI used the same wording as in our original Study 1’, but replaced “explainable” with “understandable”:

Definition: “By understandable AI we mean that an AI’s decision can be explained in non-technical terms. In other words, it is possible to know and to understand how an AI arrives at its decision.”

DV: “In this case, how important is it that the AI is understandable?”, answered on a slider measure from 1=not at all important / 2 / 3=moderately important / 4 / 5=extremely important

We fully replicate our previous findings using this adapted terminology, as reported in Studies 1B and 1C in the revised manuscript.

7.2 I found Figure 3 a bit confusing to read – was there a reason why not to split (b) into Figure 3 and (c) into Figure 4? They seem like separate results. Although given that there was no three-way interaction, I don't think (c) is very important to include.

We agree that Figure 3 in our original manuscript was not well designed and included results from Study 4' that did not significantly advance our understanding of people's attitudes towards interpretability in AI. In our revised submission, Study 4' is moved to the SI and the revised figure (which is now Figure 4) is modified such that it includes (A) the slider used in Study 3' (Study 3B in the revised manuscript) along with (B) results from this study as well as (C) results from a replication that used a between-subjects design. We think the revised figure is more informative and concise.

7.3 See comment #2 above – these results feel quite intuitive to me.

Please see our above response to your comment #2.

8. I would really appreciate more detail on why the agency and patient perspectives were tested. Was there any expectation that those perspectives would differ? If not, why were they tested? Beyond the new factor of agency vs patient, and measuring accuracy and interpretability separately, am I right in concluding that Study 4 is a replication of Study 3? I am not sure if anything new has been learned in Study 4.

These are fair points that we took to heart by (i) expanding on our motivation for differentiating between agency and patient perspective in Study 4' but also by (ii) moving Study 4' into the SI. The revised manuscript now reports a new, different Study 4, which we think extends the preceding findings in a more useful way by exploring the relationship between interpretability and accuracy more deeply.

Regarding our motivation for Study 4' (=Study 5 in the SI), we now elaborate that it was motivated by the frequent claim of interpretability in AI providing a means to the higher end of justifying machine-generated decisions (e.g., Biran & Cotton, 2017; Zhang & Dafoe, 2019). This end is conceivably more important from the perspective of a responsible *agent* who oversees the decision relative to the perspective of a *patient* who is affected by the decision and might hence lead to enhanced interpretability-requirements by agents relative to patients.

9. This was a very well-written Discussion – I enjoyed reading it!
Overall, I thought this paper was quite interesting and I hope to see more work like it from the authors in the future. I hope that some of the feedback above will be useful.

Signed,
Juliana Schroeder

REVIEWERS' COMMENTS

Reviewer #1 (Remarks to the Author):

Upon reading the reviews by the other reviewers, the replies by the authors to all comments, and the updated manuscript, I believe that the authors not only have convincingly addressed my comments and those of the other reviewers (in my opinion), but they have also gone beyond that by 1) running 4 new studies in order to deal with methodological concerns about representativeness of their samples, within-subject design, and annotations of stakes/scarcity; 2) showing that their results are robust and replicable over different variations and among different samples; 3) following proper scientific practices in pre-registration and transparency/accessibility of code/data/methods; and 4) restructuring the paper to provide a coherent story and compelling results.

Given that the authors took a very long time to resubmit this version (almost 2 years), I think it is also fair to re-evaluate the contribution and novelty of the revised manuscript within the current literature (rather than the literature when it was first submitted almost 2 years ago). Especially since research on explainable AI (and the related concepts of interpretable AI, transparent AI, the blackbox problem, etc.) have exploded in the last few years. Despite all the research progress on this topic, I believe the paper still makes a significant novel contribution in being the first to provide a thorough and robust investigation into the public attitudes towards the explainability of AI-based systems. I also think that this paper fills an important gap. The explainability of an AI system is indeed an empirical question; whether an explanation is good or bad highly depends on the perception of the receiving stakeholder, and it has an impact on other important relevant issues including trust, adoption, responsibility, and blame. Yet, quantitative user studies and public attitudes still fall short in providing a thorough and robust analysis like this one. In fact, this study is concerned with a question that should be tackled before the discussion on XAI takes place: whether, when, and to what degree do citizens (or the public) actually value explainability of AI systems in the first place?

Therefore, I recommend accepting this version.

Reviewer #3 (Remarks to the Author):

I reviewed the original paper as well as the revised paper. Overall, the authors were very thorough in addressing all of my questions and those of the other reviewers. After reading the revised paper, I felt reassured that the empirical evidence reported is now much stronger than in the original paper. The authors added several new conceptual replications that address prior concerns: Studies 1b-c replicate Study 1a with more representative samples, Study 3a examines the accuracy/interpretability trade-off, and Study 3c is a between-subjects replication of Studies 2 and 3b. The focus in the revised paper is also clearer, showing that when there are higher stakes and scarcer resources, there is a stronger preference for interpretable AI.

However, I have a few remaining thoughts and ideas that I include below:

1. There was one claim that the authors made that I did not think was clearly enough explained: that people value interpretability independent of accuracy (top of p. 16, also in the discussion of Study 3a). I don't see how the results from 3a support that claim, especially given that there was a positive association between accuracy and interpretability, so the constructs were not orthogonal. Perhaps the authors could clarify that claim.

2. This is really more of a general comment, but the findings in the paper are rather intuitive. It isn't very surprising that interpretability is more valued when the stakes are higher, or that people have a preference for accuracy when they are forced to make a trade-off between interpretability and accuracy. In their response letter, the authors argue that "the intuitiveness of this insight does not reduce its significance" – and I agree. I also think that the topic of the paper is important, and interesting. But at the same time, I'd urge them to think a little more about what is the novel aspect of these results and feature that more prominently in their introduction and conclusion.

3. Very minor but I noticed that the Study 2 question reported in the text ("how important is it that the AI is explainable?") is different from question in the Figure 2 ("how important is it that the AI is explainable, even if it is accurate?") Could the authors clarify the exact wording and be consistent in reporting it?

4. I assume this is a point for future research, but I'll just mention it in case the authors find it interesting. It strikes me that people may value interpretability either because they personally can now better understand/explain the algorithm results *or* because they think that at least someone else could understand the algorithm, even if they themselves can not. I'm curious about which of those aspects of interpretability are more appealing, or if one is seen as just as good as the other.

5. Another comment – while their hypothesized results did replicate in the between-subjects design of Study 3c, the effects were extremely small ($d=.03$, $d=.06$). As comparison, for most social science results, a d of .2 would be considered small, and a d close to 0 would be considered a null effect. So I struggle to know how to interpret those results. The paper may be served by providing clearer comparison standards to help the reader know how to interpret the strength of the effects. Good luck to the authors as they continue this interesting line of research.

Signed,

Juliana Schroeder

We responded to all remaining reviewer queries as follows:

Reviewer #1 (Remarks to the Author):

Upon reading the reviews by the other reviewers, the replies by the authors to all comments, and the updated manuscript, I believe that the authors not only have convincingly addressed my comments and those of the other reviewers (in my opinion), but they have also gone beyond that by 1) running 4 new studies in order to deal with methodological concerns about representativeness of their samples, within-subject design, and annotations of stakes/scarcity; 2) showing that their results are robust and replicable over different variations and among different samples; 3) following proper scientific practices in pre-registration and transparency/accessibility of code/data/methods; and 4) restructuring the paper to provide a coherent story and compelling results.

Given that the authors took a very long time to resubmit this version (almost 2 years), I think it is also fair to re-evaluate the contribution and novelty of the revised manuscript within the current literature (rather than the literature when it was first submitted almost 2 years ago). Especially since research on explainable AI (and the related concepts of interpretable AI, transparent AI, the blackbox problem, etc.) have exploded in the last few years. Despite all the research progress on this topic, I believe the paper still makes a significant novel contribution in being the first to provide a thorough and robust investigation into the public attitudes towards the explainability of AI-based systems. I also think that this paper fills an important gap. The explainability of an AI system is indeed an empirical question; whether an explanation is good or bad highly depends on the perception of the receiving stakeholder, and it has an impact on other important relevant issues including trust, adoption, responsibility, and blame. Yet, quantitative user studies and public attitudes still fall short in providing a thorough and robust analysis like this one. In fact, this study is concerned with a question that should be tackled before the discussion on XAI takes place: whether, when, and to what degree do citizens (or the public) actually value explainability of AI systems in the first place?

Therefore, I recommend accepting this version.

Response to Reviewer #1:

We are grateful that Reviewer #1 took the time to reassess our revised manuscript along with our responses to their and their colleagues' previous comments. At this stage, Reviewer #1 remarks that "despite all the research progress on this topic, [...] the paper still makes a significant novel contribution in being the first to provide a thorough and robust investigation into the public attitudes towards the explainability of AI-based systems" and recommends accepting the revised version.

Reviewer #3 (Remarks to the Author):

I reviewed the original paper as well as the revised paper. Overall, the authors were very thorough in addressing all of my questions and those of the other reviewers. After reading the revised paper, I felt reassured that the empirical evidence reported is now much stronger than in the original paper. The authors added several new conceptual replications that address prior concerns: Studies 1b-c replicate Study 1a with more representative samples, Study 3a examines the accuracy/interpretability trade-off, and Study 3c is a between-subjects replication of Studies 2 and 3b. The focus in the revised paper is also clearer, showing that when there are higher stakes and scarcer resources, there is a stronger preference for interpretable AI.

However, I have a few remaining thoughts and ideas that I include below:

1. There was one claim that the authors made that I did not think was clearly enough explained: that people value interpretability independent of accuracy (top of p. 16, also in the discussion of Study 3a). I don't see how the results from 3a support that claim, especially given that there was a positive association between accuracy and interpretability, so the constructs were not orthogonal. Perhaps the authors could clarify that claim.
2. This is really more of a general comment, but the findings in the paper are rather intuitive. It isn't very surprising that interpretability is more valued when the stakes are higher, or that people have a preference for accuracy when they are forced to make a trade-off between interpretability and

accuracy. In their response letter, the authors argue that “the intuitiveness of this insight does not reduce its significance” – and I agree. I also think that the topic of the paper is important, and interesting. But at the same time, I’d urge them to think a little more about what is the novel aspect of these results and feature that more prominently in their introduction and conclusion.

3. Very minor but I noticed that the Study 2 question reported in the text (“how important is it that the AI is explainable?”) is different from question in the Figure 2 (“how important is it that the AI is explainable, even if it is accurate?”) Could the authors clarify the exact wording and be consistent in reporting it?

4. I assume this is a point for future research, but I’ll just mention it in case the authors find it interesting. It strikes me that people may value interpretability either because they personally can now better understand/explain the algorithm results *or* because they think that at least someone else could understand the algorithm, even if they themselves can not. I’m curious about which of those aspects of interpretability are more appealing, or if one is seen as just as good as the other.

5. Another comment – while their hypothesized results did replicate in the between-subjects design of Study 3c, the effects were extremely small ($d=.03$, $d=.06$). As comparison, for most social science results, a d of .2 would be considered small, and a d close to 0 would be considered a null effect. So I struggle to know how to interpret those results. The paper may be served by providing clearer comparison standards to help the reader know how to interpret the strength of the effects.

Good luck to the authors as they continue this interesting line of research.

Signed,

Juliana Schroeder

Response to Reviewer #3:

Again, we are grateful for the thoughtful review of our revised manuscript by Dr Schroeder, who “felt reassured that the empirical evidence reported is now much stronger than in the original paper”. We address her remaining comments as follows:

1. There was one claim that the authors made that I did not think was clearly enough explained: that people value interpretability independent of accuracy (top of p. 16, also in the discussion of Study 3a). I don’t see how the results from 3a support that claim, especially given that there was a positive association between accuracy and interpretability, so the constructs were not orthogonal. Perhaps the authors could clarify that claim.

In view of the Reviewer’s comment, we revisited the sections discussing the results of Study 3A. We think the observation reported in the discussion of Study 3A (“Study 3A asked participants to evaluate the importance of interpretability across *independently varying levels of accuracy*”; emphasis added) is appropriate as is. Meanwhile, we agree that the claim on top of p. 16 does not follow immediately from Study 3A and hence dropped it. The revised passage now reads:

“When we explored participants’ attitudes towards interpretability without imposing such a tradeoff, we found that most participants rated interpretability as invariably important across all levels of AI accuracy, indicating they value interpretability in AI in its own right.” (p. 12)

2. This is really more of a general comment, but the findings in the paper are rather intuitive. It isn’t very surprising that interpretability is more valued when the stakes are higher, or that people have a preference for accuracy when they are forced to make a trade-off between interpretability and accuracy. In their response letter, the authors argue that “the intuitiveness of this insight does not reduce its significance” – and I agree. I also think that the topic of the paper is important, and interesting. But at the same time, I’d urge them to think a little more about what is the novel aspect of these results and feature that more prominently in their introduction and conclusion.

We thank the Reviewer for prompting us to further reflect upon this point. As pointed out by Reviewer 1, we think that the novelty and significance of our results is that we address an empirical gap by providing evidence on “whether, when, and to what degree do citizens (or the public) actually value explainability of AI systems” (quoting Reviewer 1). Furthermore, we think that not all our results are intuitive -- for instance, people’s readiness to sacrifice interpretability for accuracy in tradeoffs stands in contrast to our findings from Study 3A and also appears remarkable against the

current emphasis on interpretability in expert discussions. That said, to highlight the novelty of our contribution more prominently in our introduction, we have added the following sentence: “By testing these hypotheses among non-experts (see the Methods section for summaries of participants’ computer science knowledge), we sought to address the present lack of empirical insights about public attitudes towards AI interpretability.” (p. 4)

3. Very minor but I noticed that the Study 2 question reported in the text (“how important is it that the AI is explainable?”) is different from question in the Figure 2 (“how important is it that the AI is explainable, even if it is accurate?”) Could the authors clarify the exact wording and be consistent in reporting it?

We thank the Reviewer for pointing us to this inconsistency. The correct instructions were stated in the main text and we have corrected the caption of Figure 2 correspondingly. We also added information about a note that was displayed along with the question in Study 2, and we now point to the SI Methods which include a screenshot of the question:

“For each application and version, participants answered the question “In this case, how important is it that the AI is explainable?” using a slider ranging from “not at all important” to “extremely important”. Below the slider, we displayed a note reminding participants that “Explainable means that the AI’s decision can be explained in non-technical terms. Please consider how important it is that the AI is explainable, even if it performs accurately” (emphasis from original instructions; see SI Methods, Materials Study 2).” (p. 7 f.)

4. I assume this is a point for future research, but I’ll just mention it in case the authors find it interesting. It strikes me that people may value interpretability either because they personally can now better understand/explain the algorithm results *or* because they think that at least someone else could understand the algorithm, even if they themselves can not. I’m curious about which of those aspects of interpretability are more appealing, or if one is seen as just as good as the other.

Especially in regard to questions of AI ethics and governance, it would be intriguing to explore and disentangle these potential motives – something that we will keep in mind for our future work. We thank the Reviewer for sharing this inspiring thought.

5. Another comment – while their hypothesized results did replicate in the between-subjects design of Study 3c, the effects were extremely small ($d=.03$, $d=.06$). As comparison, for most social science results, a d of .2 would be considered small, and a d close to 0 would be considered a null effect. So I struggle to know how to interpret those results. The paper may be served by providing clearer comparison standards to help the reader know how to interpret the strength of the effects.

We agree that the observed effect sizes in Study 3C are extremely small. In fact, in the previously submitted version of our manuscript, we explicitly mentioned the small effect sizes observed and proposed that they might point towards people’s sensitivity to variations in stakes and scarcity being dependent on the salience of the variation in the two attributes. Endorsing the Reviewer’s comment, we made further changes to this passage to help the reader interpret the small effect sizes observed: “However, effect sizes relative to Study 3B were extremely small. This suggests that people’s sensitivity to stakes and scarcity is dependent on the salience of variation in the two attributes, which was higher in the within-subjects design than the between-subjects design. Indeed, as we report in Study 3D in the SI, when we ran an additional experiment that reduced the salience of variation of the two attributes to a minimum, by not even mentioning their range, only the main effect for stakes remained significant ($p < .001$) whereas the effect for scarcity was no longer significant ($p = .136$).” (p. 11)